# Ampere-level current density ammonia electrochemical synthesis using CuCo nanosheets simulating nitrite reductase bifunctional nature

Jia-Yi Fang [1,4], Qi-Zheng Zheng [1,4], Yao-Yin Lou [1,2] ✉, Kuang-Min Zhao [1], Sheng-Nan Hu [1], Guang Li[1], Ouardia Akdim[3], Xiao-Yang Huang[1,3] & Shi-Gang Sun [1] ✉

The development of electrocatalysts capable of efficient reduction of nitrate ($NO_3^-$) to ammonia ($NH_3$) is drawing increasing interest for the sake of low carbon emission and environmental protection. Herein, we present a CuCo bimetallic catalyst able to imitate the bifunctional nature of copper-type nitrite reductase, which could easily remove $NO_2^-$ via the collaboration of two active centers. Indeed, Co acts as an electron/proton donating center, while Cu facilitates $NO_x^-$ adsorption/association. The bio-inspired CuCo nanosheet electrocatalyst delivers a $100 \pm 1\%$ Faradaic efficiency at an ampere-level current density of 1035 mA cm$^{-2}$ at −0.2 V *vs.* Reversible Hydrogen Electrode. The $NH_3$ production rate reaches a high activity of 4.8 mmol cm$^{-2}$ h$^{-1}$ (960 mmol g$_{cat}^{-1}$ h$^{-1}$). A mechanistic study, using electrochemical in situ Fourier transform infrared spectroscopy and shell-isolated nanoparticle enhanced Raman spectroscopy, reveals a strong synergy between Cu and Co, with Co sites promoting the hydrogenation of $NO_3^-$ to $NH_3$ via adsorbed *H species. The well-modulated coverage of adsorbed *H and *$NO_3$ led simultaneously to high $NH_3$ selectivity and yield.

Nitrate anions ($NO_3^-$), widely present in industrial and agricultural wastewater, pose a real potential threat to human health and ecological balance, especially their incomplete conversion into nitrites ($NO_2^-$) which are thought to be cancerogenic by inducing liver damage and methaemoglobinaemia[1]. The conventional biological treatments for $NO_3^-$ removal into nitrogen ($N_2$) gas, involving nitrification and denitrification processes, are energy intensive (-11.7 to 12.5 kWh kg N$^{-1}$)[2]. Actually, the reduction of $NO_3^-$ into $NH_3$ has become of great interest from an industrial point of view since $NH_3$ is a highly important industrial chemical for the synthesis of pharmaceuticals, fertilizers, dyes, plastic, etc.[3], and is also considered for hydrogen storage/release as a carbon-free hydrogen carrier[4]. To date, the industrial synthesis of $NH_3$ relies heavily on the non-sustainable and eco-unfriendly Haber−Bosch route, which requires harsh conditions i.e., high temperature (400−600 °C) and high pressure (200−350 atm), and heavily relies on fossil energy[3]. The total amount of $CO_2$ produced during the Haber−Bosch process accounts for roughly 1.2% of the global annual $CO_2$ emissions, more than any other industrial chemicals synthesis[5].

Electrocatalytic reduction of $NO_3^-$ into $NH_3$, powered by green energy, has drawn increasing attention and is considered as a

[1]State Key Laboratory of Physical Chemistry of Solid Surfaces, Department of Chemistry, College of Chemistry and Chemical Engineering, Xiamen University, Xiamen 361005, China. [2]School of Chemical and Environmental Engineering, College of Chemistry, Chemical Engineering and Materials Science, Soochow University, Suzhou, Jiangsu 215123, China. [3]Cardiff Catalysis Institute, School of Chemistry, Cardiff University, Main Building, Park Place, Cardiff CF10 3AT, UK. [4]These authors contributed equally: Jia-Yi Fang, Qi-Zheng Zheng ✉e-mail: yylou@suda.edu.cn; sgsun@xmu.edu.cn

sustainable complementary process to the Haber–Bosch process[2,6,7], since this technology can simultaneously meet the 21st session of the Conference of the Parties (COP) two-degree scenario (2DS) target for $NH_3$ and protect the environment from the eutrophic water pollution. The rational design of novel electrocatalysts with both high activity and selectivity is crucial for reducing $NO_3^-$ ($NO_3^-RR$) and achieving large-scale applications, and satisfying high industrial demands. Since Nature has developed sophisticated and efficient machinery such as enzymes[8,9], it is interesting to design catalysts by studying the mechanism of enzymatic reduction. Indeed, biocatalytic reduction of $NO_3^-$ to $NH_3$ widely exists inside many microorganisms, where $NO_3^-$ ions are firstly reduced into $NO_2^-$ by $NO_3^-$ reductases that accept electrons from quinone[10]. The generated $NO_2^-$ is further converted to $NH_3$ by nitrite reductases (NIRs) using quinone as electron donors. Among various NIRs, the Cu-type NIRs (Cu-NIRs) found in Rhizobium is one of the most important enzymes for $N_2$ fixation. Cu-NIRs are trimeric proteins composed of 3 identical subunits, and each monomer has two types of Cu atomic active centers, acting as electron-donating centers (T1Cu) and catalytic centers (T2Cu), respectively[10]. The reported mechanism proposes that $*NO_2^-$ (where * denotes an adsorbed specie) is bound to the T2Cu in a bidentate form via two oxygen atoms. Due to the electrons transfer from T1Cu to T2Cu, the T2Cu oxidation state decreases from (II) to (I), facilitating $*NO_2^-$ association with T2Cu in a bridging nitro binding form. Meanwhile, the aspartate acid besides the T2Cu provides proton to one oxygen and extends the N-O bond length, leading to the breaking of the N-O bond[11]. According to the mechanism of $NO_2^-$ reduction on T2Cu and T1Cu, it can be speculated that moderate affinity with $NO_3^-$, protons availability and electrons provision are the key factors for the high efficiency in $NO_3^-RR$ to $NH_3$. Jimmy C. Yu et al.[12] proved that hydrogen adsorption (*H), with moderate adsorption energy on catalysts, can behave as an important reactive species for the hydrogenation of $NO_x$ intermediates to $NH_3$ whilst suppressing the hydrogen evolution reaction (HER) competition. Matthew J. Liu et al.[13] also found that the $NO_3^-RR$ activity and product selectivity highly depended on the *H coverage on the titanium surface at different potentials with the increase of $NO_3^-RR$ overpotential.

Very recently, Schuhmann and his colleagues[14] reported a tandem mechanism for $PO_4^{3-}$-modified CuCo binary metal sulfides. They proposed that $NO_2^-$ intermediates were preferentially formed on Cu-based phases followed by splitting over nearby Co-based phases; however, the role of *H during $NO_3^-RR$ was unclear and the effect of the electronic structure of alloyed catalysts on the adsorption of reaction intermediates lacked further discussions in the proposed mechanism[14]. In this study, inspired by the bifunctional nature of the Cu-NIRs, we prepared CuCo alloy nanosheets via a one-step electrodeposition route. The CuCo alloy could well mimic the behavior of the two catalytic centers in Cu-NIRs. The Co species could efficiently provide the electrons and generate the hydrogen protons to the nearby Cu species with high adsorption of $NO_3^-$ and its derivatives. The as-prepared CuCo nanosheet exhibited superior catalytic performances, as (1) the lowest η of 290 mV for ammonia production (at 0.4 V vs. reversible hydrogen electrode (RHE), all electrode potentials mentioned below were provided with respect to RHE unless specially stated); (2) an ampere-level current density of 1035 mA cm$^{-2}$ with a 100% Faradaic efficiency for $NH_3$ generation at an overpotential of 890 mV, and a corresponding Yield$_{NH_3}$ of 4.8 mmol cm$^{-2}$ h$^{-1}$; (3) a wide potential window (300 mV, from −0.1 to −0.4 V) for $NH_3$ generation with >90% Faradaic efficiency (FE), which is one of the most advanced catalysts for $NO_3^-RR$ so far. Electrochemical in situ Fourier transform infrared spectroscopy (FTIR) and in situ shell-isolated nanoparticle enhanced Raman spectroscopy (SHINERS) associated with density functional theory (DFT) calculations were conducted to clarify the pathways and mechanisms of $NO_3^-RR$, with the aim to contribute to subsequent catalyst optimization and scaling up.

## Results

### Preparation and characterization of CuCo bimetallic electrocatalysts

CuCo bimetallic materials were synthesized by co-electrodeposition of Cu and Co on Ni foams' surfaces (Fig. 1a and Method section). Ni foam is widely used as a supporting substrate for nanostructured electrocatalysts due to its smooth surface and good conductivity, benefiting the electrodeposition by an efficient electron transfer[15–17]. Meanwhile, Ni was proved as a relatively inert material for $NO_3^-RR$[18,19], without affecting the CuCo catalysts' performance. The Cu/Co molar ratio was determined using an inductively coupled plasma-optical emission spectrometer (ICP-OES) (Supplementary Table S1). The catalyst with a Cu/Co molar ratio of ca. 50/50 was named $Cu_{50}Co_{50}$ and was considered as a reference in this study. The alloying of Cu and Co in $Cu_{50}Co_{50}$ was confirmed by X-ray diffraction (XRD) and high-resolution transmission electron microscopy (HRTEM). Indeed, in the diffractogram (Fig. 1b and Supplementary Fig. S1) shifts in the Cu (111) and Cu (200) diffraction peaks toward higher degrees were observed after the addition of Co to Cu, which were attributed to the shrinkage of the lattice spacing caused by the partial alloying of Co atom with a smaller diameter compared to Cu atom[20]. Furthermore, the lattice spacing of the Cu (111) plane contracted to 0.208 nm after alloying Cu with Co (Fig. 1c), when it was 0.212 nm for the Cu (111) plane for the pure Cu catalyst (Supplementary Fig. S2a)[15]. Scanning electron microscopy (SEM) was applied to examine the morphologies of Cu, Co and $Cu_xCo_y$ catalysts and showed micro-pines structure for all the catalysts (Supplementary Fig. S3a–j). At the nanoscale level, small bump structures on the surface of the micro-pines were observed on pure Cu (Supplementary Fig. S3a, b). After the incorporation of Co, a nanosheet structure emerged on the micro-pine's surface of the $Cu_{50}Co_{50}$ catalyst (Fig. 1d and Supplementary Fig. S3e, f), very similar to the structure of pure Co (Supplementary Fig. S3i, j). The thickness of the $Cu_{50}Co_{50}$ nanosheet was evaluated by atomic force microscopy (AFM) and was around 10 nm (Fig. 1e). Besides, the EDS mapping analysis disclosed an even distribution of Cu and Co in $Cu_{50}Co_{50}$ (Fig. 1f).

The electronic properties of the $Cu_{50}Co_{50}$ nanosheet were explored by X-ray photoelectron spectroscopy (XPS). $Cu^{2+}$ 2p peaks were observed in the high-resolution Cu 2p spectra (Fig. 1g), and was due to the partial oxidation of the alloy's surface exposed to air, and the same observation was made for Co (Fig. 1h). The decrease of the Cu 2p binding energy, compared with pure Cu (Supplementary Fig. S4a) and the notable increase of the Co 2p binding energy, compared with pure Co (Supplementary Fig. S4b), revealed a redistribution of the electrons between Cu and Co after their alloying[21], leading to the movement of the d band towards the Fermi level[22] comparing with the pure Cu catalyst. X-ray absorption spectroscopy (XAS) analysis was also performed to check the redistribution of electrons between Cu and Co. Extended X-ray absorption fine structure (XANES) spectra depicted a negative shift of the absorption edge position for the Cu K-edge after interacting with Co (indicated by the red arrow in the inset) (Supplementary Fig. S5), illustrating the electron density transfer from Co to Cu[23]. In addition, according to the Bader charge analysis (Supplementary Fig. S6), compared with monometallic Cu and Co, a charge redistribution on CuCo(111) was observed, where the Cu center displayed a higher electrons density compared to the Co center (Supplementary Fig. S7)[24]. The correlation between electrons redistribution within Cu and Co and the adsorption energy of *H, $*NO_3$ and the $*NO_x$ intermediates[25] are discussed further in the mechanistic study section.

### Electrochemical activity and kinetics of $NO_3^-RR$

$NO_3^-RR$ was first investigated by linear sweep voltammetry (LSV) at a low scan rate of 1 mV s$^{-1}$ on all the prepared electrocatalysts. The reduction of $NO_x$ species contributed mainly to the current density (solid curve), indicating a high catalytic activity toward $NO_3^-RR$ for all the catalysts (Fig. 2a). The substrate of Ni foam was inactive for $NO_3^-RR$

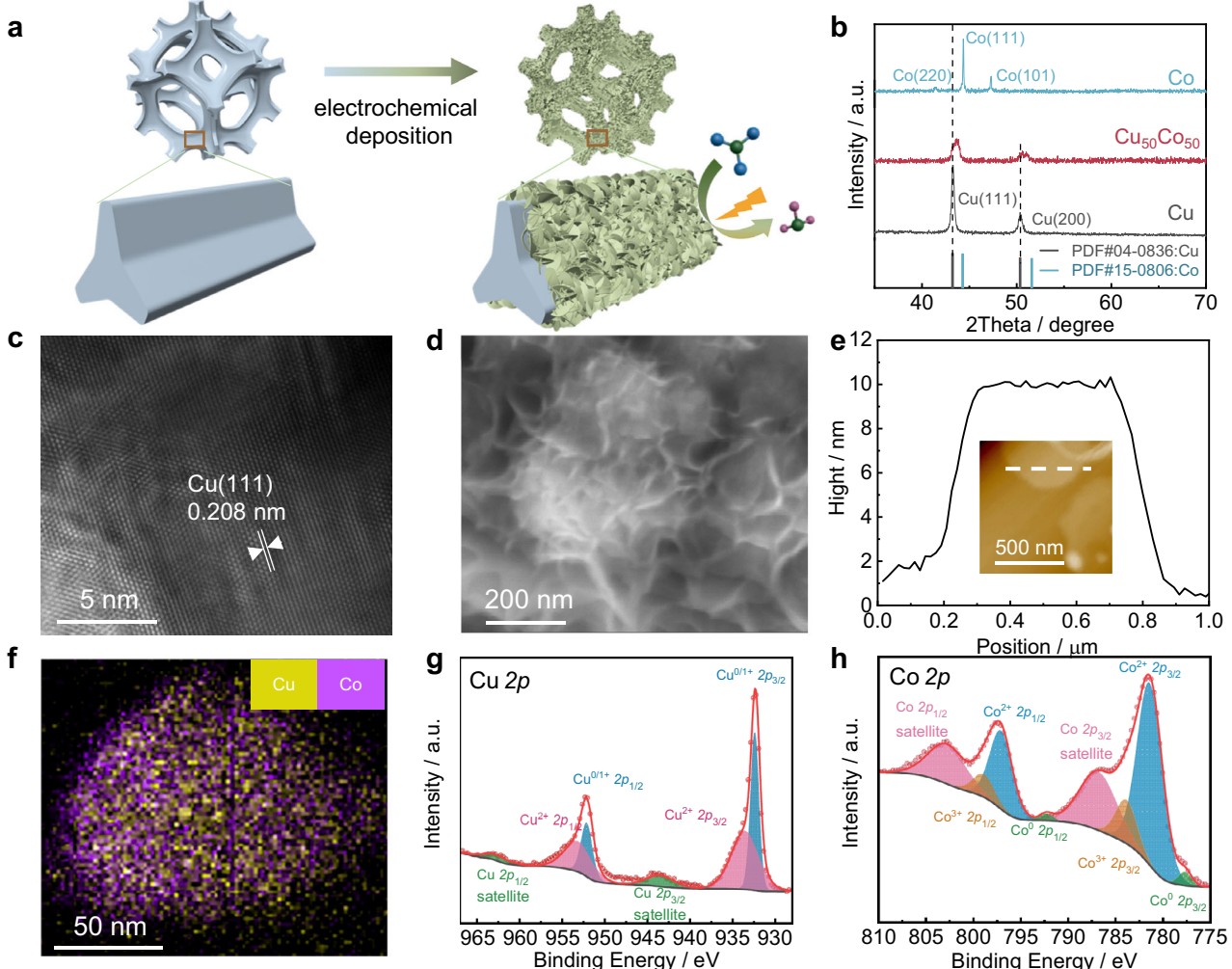

**Fig. 1 | Preparation strategy and characterization of catalysts. a** Schematic diagram of CuCo alloy electrodeposition on nickel foam's surface. **b** XRD spectra of $Cu_{50}Co_{50}$, Cu, and Co. HRTEM image (**c**), SEM image (**d**), linear topography profiles from AFM images (**e**), and EDS mapping analysis (**f**) of $Cu_{50}Co_{50}$. XPS peaks spectra of Cu 2p (**g**) and Co 2p (**h**) of $Cu_{50}Co_{50}$.

compared with the electrodeposited catalysts (Supplementary Figs. S8, S9). The overpotential η at 10 mA cm⁻² (denoted $\eta_{@10mAcm^{-2}}$, η = $E^{o}$ − E, where E is the potential with a current density of 10 mA cm⁻², $E^{o}$ = 0.69 V), where $NO_3^-$RR is under kinetic control, was used as a criterion to compare the catalysts' activity. $Cu_{50}Co_{50}$ and Cu exhibited a lower energy barrier for $NO_3^-$RR, with $\eta_{@10mAcm^{-2}}$ of 498 and 503 mV, respectively, compared to Co, that displayed a $\eta_{@10mAcm^{-2}}$ of 690 mV. In fact, two reduction current peaks (peak S1 and S2) were observed in the curves of Cu and $Cu_{50}Co_{50}$. At the initial stage of the reaction, Cu and $Cu_{50}Co_{50}$ seemed to display a similar behavior toward $NO_3^-$RR, suggesting the important role of Cu at this stage. According to previous studies[26,27], the peak S1 near 0.08 V was assigned to $^*NO_3^-$ (adsorbed $NO_3^-$) reduction into $^*NO_2^-$ (1) following a 2-electrons transfer process, while the peak S2 was allocated to the $^*NO_2^-$ reduction into $^*NH_3$ (2) following a 6-electrons transfer process. In the peak S2 region, $Cu_{50}Co_{50}$ displayed a positive potential shift of 67 mV, compared to Cu, suggesting a significant synergy between Co and Cu and a drop of the barrier energy of $^*NO_2^-$ reduction to $^*NH_3$, as the applied potential increases.

$$^*NO_3^- + 2e^- + H_2O \rightarrow {}^*NO_2^- + 2OH^- \tag{1}$$

$$^*NO_2^- + 6e^- + 5H_2O \rightarrow {}^*NH_3 + 7OH^- \tag{2}$$

In the aim to investigate the reaction routes for all the catalysts, the number of electrons transferred (n) during the $NO_3^-$ reduction reaction was estimated from the slope of the Koutecký–Levich (K–L) plots (Supplementary Fig. S10). For the Cu catalyst (Supplementary Fig. S11a), n was 2 at −0.1 V and a quasi-first-order reaction relationship between j and the $NO_3^-$ concentration was obtained (Supplementary Fig. S12), validating the rate-determinate step (RDS) as the reduction route described by Eq. (1). The n value increased to 6 at −0.25 V, validating the reduction route described in Eq. (2). In comparison, a direct 8-electrons transfer process was observed on $Cu_{50}Co_{50}$ in the potential region between the peaks S1 and S2 (Fig. 2b), revealing a strong alloying effect promoting simultaneously both routes. Though an 8-electrons transfer process also occurred on the pure Co catalyst at a far less negative potential of ca. −0.45 V (Supplementary Fig. S11b), i.e., at a higher barrier energy.

In the potential region of peak S1, the Tafel slopes derived from the j-E curves (Fig. 2c), for $Cu_{50}Co_{50}$ was 205.75 mV decade⁻¹, which was slightly lower than Cu (232.43 mV decades⁻¹), indicating that the addition of Co could promote the electrons transfer over the catalyst/electrolyte interface during $^*NO_3^-$ reduction to $^*NO_2^-$. This result was supported by EIS data (Supplementary Fig. S13), demonstrating a smaller charge-transfer resistance on $Cu_{50}Co_{50}$ compared to Cu (3.51 vs. 3.81 Ω). The Co catalyst had minor electrocatalytic activity for $NO_3^-$ reduction at this relatively positive working potential, so its Tafel slop

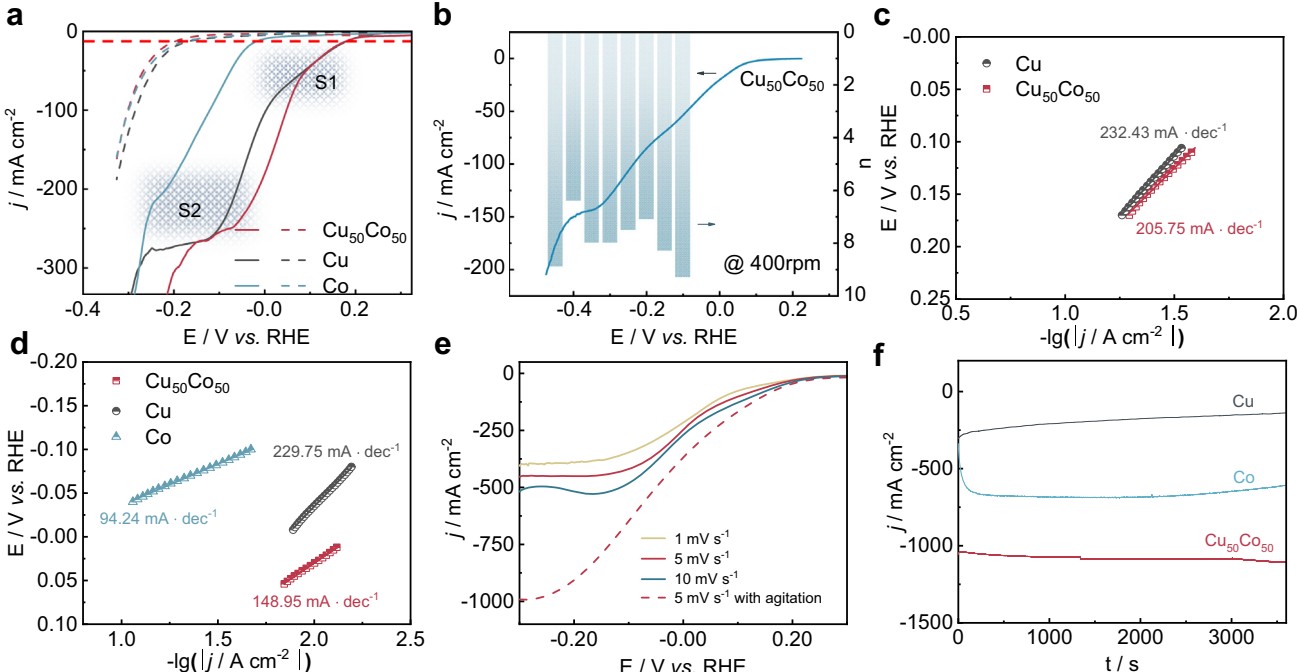

**Fig. 2 | Electrochemical responses of Cu$_{50}$Co$_{50}$, pure Cu, and pure Co catalysts.** **a** *j-E* curve (80% i*R* corrected) over Cu$_{50}$Co$_{50}$, pure Cu, and pure Co modified Ni foams (catalysts loading was 5 mg cm$^{-2}$) in 1 M KOH solution containing 100 mM KNO$_3$ (solid lines) or in the absence of KNO$_3$ (dotted line) at a scan rate of 1 mV s$^{-1}$ (the red dash line presenting the *j* of 10 mA cm$^{-2}$, the shading S1 and S2 presenting the peak around 0.2 to 0.05 V and 0.05 to −0.15 V, respectively). **b** *j-E* curve (80% i*R* corrected) at 400 rpm and electron transfer numbers at different potentials calculated by the K–L equation for Cu$_{50}$Co$_{50}$ on RDE in 100 mM KNO$_3$ + 1 M KOH

electrolyte at a scan rate of 10 mV s$^{-1}$ (catalysts loading was 0.25 mg cm$^{-2}$). Tafel slopes in the potential range of peak S1 (**c**) S2 (**d**). **e** *j-E* curves over Cu$_{50}$Co$_{50}$ modified Ni foam in 1 M KOH solution containing 100 mM KNO$_3$ at different scan rates without agitation (solid line) and at a scan rate of 5 mV s$^{-1}$ with agitation (catalysts loading was 5 mg cm$^{-2}$). **f** Time-dependent current density curves over Cu$_{50}$Co$_{50}$, Cu, Co modified Ni foam at −0.2 V with a magnetic stirring speed of 1000 rpm (catalysts loading was 5 mg cm$^{-2}$).

in this potential region was not determined. In the potential region of peak S2, the Tafel slope (Fig. 2d) of Cu$_{50}$Co$_{50}$ (148.95 mV decades$^{-1}$) was significantly lower than Cu (229.75 mV decades$^{-1}$); this was explained by an electron redistribution between Cu and Co over Cu$_{50}$Co$_{50}$ and was emphasized in the characterization section. The Co catalyst displayed the lowest Tafel slope value of 94.24 mV decades$^{-1}$, but at a more negative potential, i.e., 95 mV shift, than Cu$_{50}$Co$_{50}$. This latter result suggested that after crossing a high energy barrier, Co displayed better kinetics' performances toward the NO$_3^-$RR for NH$_3$ production compared to the Cu-based catalysts. Alloying Co to Cu implemented a faster electron transfer rate and improved kinetics' performances toward the NO$_3^-$RR. In addition to being affected by the properties of the catalysts, the NO$_3^-$RR was also diffusion-controlled (Fig. 2e). To further evaluate the NO$_3^-$RR activity of the catalysts under steady-state conditions, potentiostatic electrolytic reduction of NO$_3^-$ in a homemade H-type electrolytic cell (Supplementary Fig. S14) was carried out by electrodepositing the catalysts on Ni foams' surfaces. A magnetic stirring rate of 1000 rpm was used to minimize the effect of diffusion and fresh electrolyte solution was constantly supplemented to maintain a constant NO$_3^-$ ion concentration. In these conditions, the current densities obtained on the Cu$_{50}$Co$_{50}$ catalyst reached an Ampere level at −0.2 V *vs.* RHE (Fig. 2f).

NH$_3$ and NO$_2^-$ were quantitatively detected by the Nessler Reagent method and ion chromatography, respectively (Supplementary Fig. S15)[28,29]. The gas products were analyzed during the reaction by gas chromatography and online electrochemical mass spectrometry (OEMS)[30]. There was no N$_2$ production detected and the amount of the stripped NH$_3$ in this work was negligible. (Supplementary Fig. S16) In order to figure out the effect of Co content on the NO$_3^-$RR activity, CuCo bimetallic catalysts with different Cu/Co ratios were prepared, i.e., Cu$_{65}$Co$_{35}$ and Cu$_{15}$Co$_{85}$. As shown in Fig. 3a, when a potential of 0 V

was applied, the Faraday efficiency of NH$_3$ (FE$_{NH_3}$) and NO$_2^-$ (FE$_{NO_2^-}$) on pure Cu were 32% and 60%, respectively. The corresponding molar ratio of NO$_2^-$ to NH$_3$ was seven times higher (Fig. 3c), suggesting an accumulation of NO$_2^-$ due to the low kinetic of *NO$_2^-$ hydrogenation to *NH$_3$ on the Cu surface. With the addition of Co to Cu, both the FE$_{NH_3}$ and the geometrically normalized current density of NH$_3$ production ($j_{NH_3}$) increased significantly (Fig. 3a, b). A volcano shape was observed and the highest $j_{NH_3}$ was obtained on Cu$_{50}$Co$_{50}$ (i.e., 347 mA cm$^{-2}$ $j_{NH_3}$ and 88% FE$_{NH_3}$), and was ten times higher than monometallic Cu (i.e., 34 mA cm$^{-2}$ and 32% FE$_{NH_3}$), and nearly 17 times higher than monometallic Co catalyst (i.e., 21 mA cm$^{-2}$ and 84% FE$_{NH_3}$). Furthermore, the molar ratio of NO$_2^-$ to NH$_3$ exhibited a dramatic drop from 7 to 0.6 as the Co content was raised from 0 to 100% (Fig. 3c). This observation suggested that the incorporation of Co enhanced further the *NO$_2^-$ hydrogenation into NH$_3$. Moreover, when the ratio of Co was increased by over 50%, the FE$_{NH_3}$ was remained constant, but the $j_{NH_3}$ declined dramatically, indicating that a moderate Cu/Co ratio was important to maintain high catalytic performance for NH$_3$ production. The electrochemically active surface area (ECSA) was also measured (Supplementary Figs. S17f, S18b), and all the catalysts had comparable ECSA. The maximum ECSA normalized current density for NH$_3$ production ($j_{NH_3(ECSA)}$) was obtained on Cu$_{50}$Co$_{50}$ (Supplementary Fig. S19), indicating that this catalyst had the highest intrinsic activity for NH$_3$ production.

To clarify the possible $^{14}$N pollution, the electrolysis in the electrolyte free of NO$_3^-$ ions was performed and little NH$_3$ was produced (Supplementary Fig. S20). $j_{NH_3}$ in the solution with NO$_3^-$ ions was over 150-fold higher than that in the electrolyte free of NO$_3^-$. The content of $^{14}$NH$_4^+$ was also analyzed by $^1$H NMR test[29,31] and was *ca.* 4.62 mmol cm$^{-2}$ h$^{-1}$. The obtained results were close to the one from the spectrophotometric analysis (4.75 mmol cm$^{-2}$ h$^{-1}$) (Supplementary

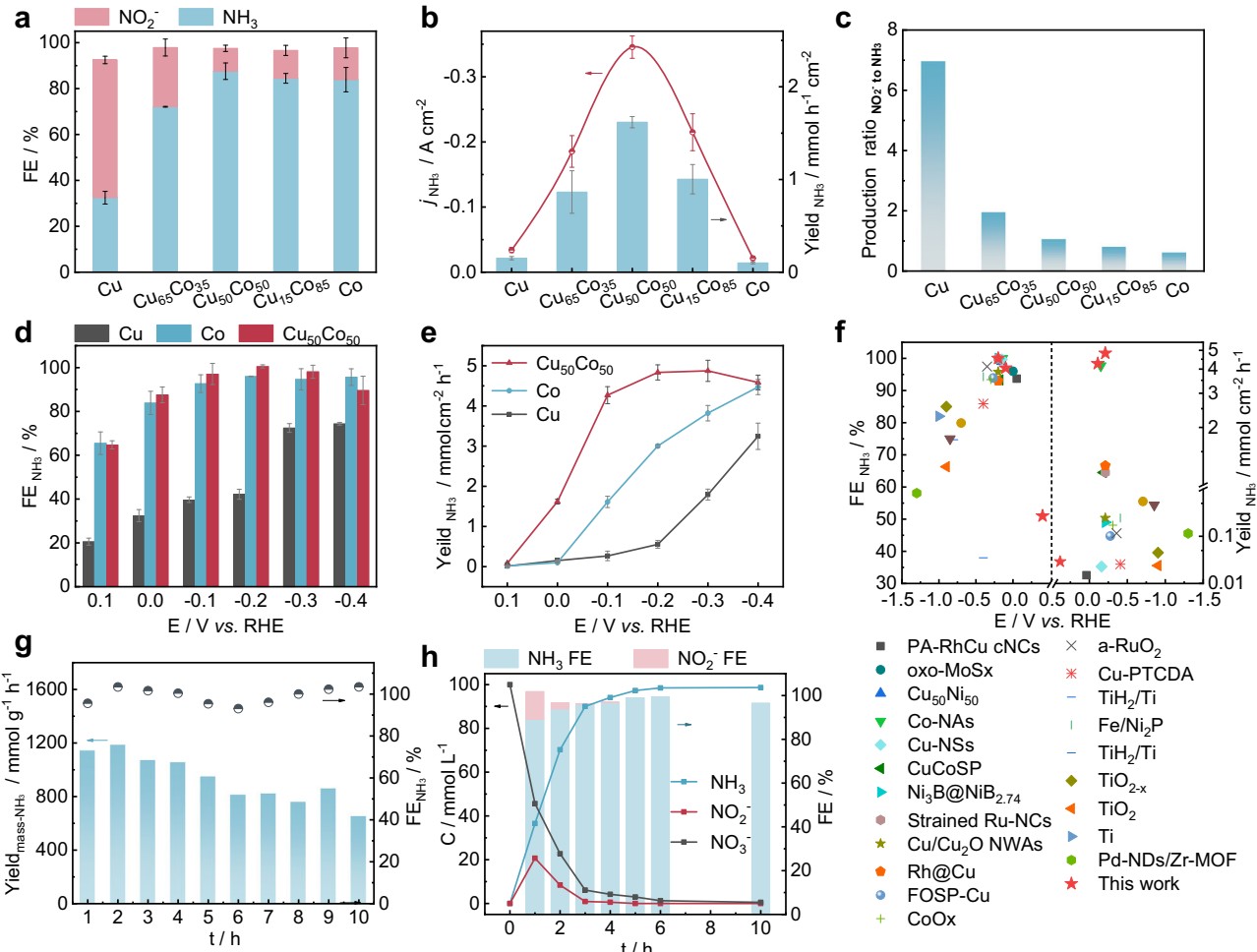

**Fig. 3 | Electrochemical performance of catalysts.** $FE_{NH_3}$ and $FE_{NO_2^-}$ of $NO_3^-$RR (**a**), bias-current density and products yield for $NH_3$ (**b**), and the ratio of $NO_2^-$ to $NH_3$ generated (**c**) for different Cu/Co ratio at 0 V in 100 mM $KNO_3$ + 1 M KOH electrolyte (catalysts loading was 5 mg cm$^{-2}$). $FE_{NH_3}$ (**d**) and $NH_3$ product yield (**e**) at different electrode potentials on $Cu_{50}Co_{50}$, pure Cu and pure Co catalysts modified Ni foam (catalysts loading was 5 mg cm$^{-2}$). Comparison of the electrocatalytic $NO_3^-$ RR performances of $Cu_{50}Co_{50}$ modified Ni foam with other extensively reported electrocatalysts (**f**). $FE_{NH_3}$ and Yield$_{mass-NH_3}$ on $Cu_{50}Co_{50}$/Ni foam under the applied potential of −0.2 V during 10 periods of 1 h electrocatalytic $NO_3^-$RR (**g**) (catalysts loading was 5 mg cm$^{-2}$). The time-dependent concentration of $NO_3^-$, $NO_2^-$ and $NH_3$ and corresponding FE over $Cu_{50}Co_{50}$ modified Ni foam at −0.1 V (**h**) (catalysts loading was 5 mg cm$^{-2}$). Error bars represent the standard deviations calculated from three independent measurements.

Fig. S21d), and confirmed the reliability of the quantification methods used in this work. The typical two peaks of $^{15}NH_4^+$ after the electrolysis of $^{15}NO_3^-$ also suggested that the $NH_3$ product indeed came from the electrocatalytic reduction of $NO_3^-$ (Supplementary Fig. S21e).

It is known that the applied potential influences the products' selectivity[13], so we investigated the effect of the applied potential toward the $NO_3^-$RR (Fig. 3d, e and Supplementary Fig. S22). The $FE_{NH_3}$ was about 51% at a η of 290 mV (at 0.4 V) on $Cu_{50}Co_{50}$. The η was comparatively much lower than that of the state-of-the-art catalysts reported in the literature (Fig. 3f and Supplementary Table S2). When the η reached 590 mV (at 0.1 V), the $FE_{NH_3}$ got about 65% on $Cu_{50}Co_{50}$ (Fig. 3d and Supplementary Fig. S23). In comparison, the $FE_{NH_3}$ was around 21% on Cu at 0.1 V (Fig. 3d), with $NO_2^-$ as the main product. As the electrode potential shifted negatively, the intermediate $NO_2^-$ was rapidly reduced (Supplementary Fig. S23). At −0.2 V, the $FE_{NH_3}$ on $Cu_{50}Co_{50}$ achieved 100 ± 1%, and $j_{NH_3}$ reached 1035 mA cm$^{-2}$ (Supplementary Fig. S24b), corresponding to an $NH_3$ production rate of 4.8 mmol cm$^{-2}$ h$^{-1}$ that is about two and eight times higher than the ones obtained on monometallic Co and Cu (Fig. 3e and Supplementary Fig. S25), respectively. Based on the charge consumed during the CuCo electrodeposition on the Ni foam substrate, the mass activity of $NH_3$ yield (Yield$_{mass-NH_3}$) on $Cu_{50}Co_{50}$ was estimated roughly to

960 mmol g$_{cat}^{-1}$ h$^{-1}$ at 0.2 V. The obtained Yield$_{mass-NH_3}$ was slightly underestimated since hydrogen evolution was observed during CuCo electrodeposition. It should be noted that the production of $NH_3$ from electrocatalytic $NO_3^-$RR is currently at a laboratory scale. More follow-up pilot tests and scale-up work are required to meet the industrial demands. Based on the preliminary calculations, we consider that our work inspired by the bifunctional nature of nitrite reductase, provides a new expectation and shows a great prospect, and after further development could compete with the well-established Haber−Bosch process[12,32] which currently shows a Yield$_{mass-NH_3}$ of ca. 200 mmol g$_{cat}^{-1}$ h$^{-1}$ at industrial scale. The potential window for $FE_{NH_3}$ above 90% was wide and ranged from −0.1 to −0.4 V (Fig. 3d). Ten cycles of electrolysis at constant potential were performed to check the stability of $Cu_{50}Co_{50}$ (Fig. 3g) and the results displayed a stable $FE_{NH_3}$ exceeding 90% over the cycles. According to the SEM analysis, the decay of the yield for $NH_3$ would be due to the nanosheet agglomeration after the consecutive recycling tests (Supplementary Fig. S26a, b). Furthermore, XRD (Supplementary Fig. S26c) and XPS (Supplementary Fig. S26d, e) analysis was also performed on the samples after the consecutive recycling tests, and the results demonstrated negligible changes in the chemical compositions and oxidation states, which confirmed the excellent stability of the catalyst.

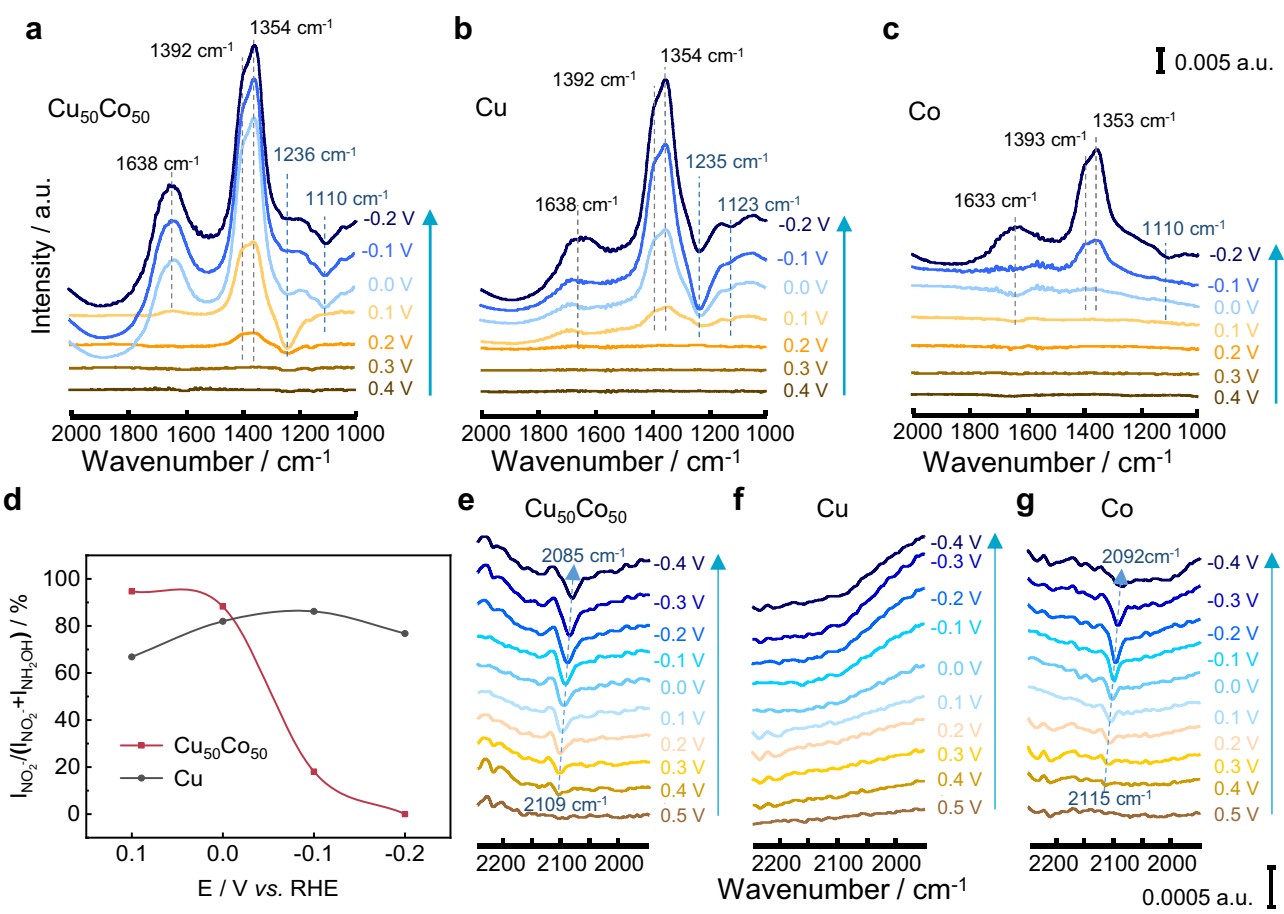

**Fig. 4 | Electrochemical in situ FTIR spectra.** Electrochemical thin-layer in situ FTIR spectra of $NO_3^-$RR on $Cu_{50}Co_{50}$ (a), Cu (b), and Co (c) in 100 mM $KNO_3$ + 1 M KOH. **d** $\frac{I_{NO_2^-}}{I_{NH_2OH} + I_{NO_2^-}}$ ratio at different electrode potentials. ATR-FTIR spectra on $Cu_{50}Co_{50}$ (e), Cu (f), and Co (g) in 1 M KOH.

The concentration of $NO_3^-$ in real wastewater can vary from 0.88 mmol $L^{-1}$ to 1.95 mol $L^{-1}$ [33]. Therefore, $NO_3^-$RR on $Cu_{50}Co_{50}$ was performed at a wide range of $NO_3^-$ concentrations (1–100 mmol $L^{-1}$). The $FE_{NH_3}$ was maintained above 95% in the whole range of $NO_3^-$ concentrations (Supplementary Fig. S27) at −0.1 V. The $NO_3^-$RR in neutral conditions was carried out on $Cu_{50}Co_{50}$ catalysts at −0.2 V in an electrolyte solution of 0.1 M $KNO_3$ + 0.5 M $K_2SO_4$. The current density of $NO_3^-$RR on $Cu_{50}Co_{50}$ in a neutral condition was higher than the one on monometallic Cu and Co, and the $FE_{NH_3}$ over $Cu_{50}Co_{50}$ was more than 90% (Supplementary Fig. S28). These experiments demonstrated the excellent property of $Cu_{50}Co_{50}$ toward $NO_3^-$ recovery in various environmental wastewater systems. In batch conditions with an initial nitrate's concentration of 100 mmol $L^{-1}$ (*ca.* 6200 ppm) at a reduction potential of −0.1 V and after 10 h, the nitrate's removal efficiency reached 99.5%, with a $FE_{NH_3}$ of 96% (Fig. 3h). The remaining $NO_3^-$ in the solution was 31 ppm which was much lower than the limitations fixed by the World Health Organization for drinking water, (i.e., 50 ppm)[34]. Several processes can be then considered for further extracting $NH_3$, such as air stripping, ion exchange, struvite precipitation, etc.[35].

Electrochemical in situ FTIR, SHINERS and DFT calculations were conducted to elucidate the reaction mechanism as well as the origin of the different activities observed between the catalysts.

### Electrochemical in situ FTIR analysis of $NO_3^-$RR
The electrochemical thin-layer in situ FTIR can track intermediates in solution within the thin-layer (thickness around 10 μm) between the electrode and IR window and species adsorbed on the electrode surface[36]. A reference spectrum ($R_{Ref}$) at reference potential ($E_R$, 0.4 V)

was firstly acquired, and then the potential was stepped to studied potentials ($E_S$) and to collect working spectra ($R_S$). The resulting spectra were represented as relative changes in the reflectance: ΔR/R = ($R_S$-$R_{ref}$)/$R_{Ref}$. As a result, the downward band in the resulting spectra indicated the formation of $NO_3^-$ intermediates at $E_S$, while the upward band referred to the consumption of $NO_3^-$. The FTIR peaks observed on $Cu_{50}Co_{50}$, Cu, and Co are compiled in Supplementary Table S3. As illustrated in Fig. 4, in the potential range from 0.4 to −0.2 V, the absorption bands were assigned to intermediates present in the electrolyte, since the wavenumbers of all the absorption bands were independent of the working potential[37]. In Fig. 4a, five obvious absorption bands appeared in the infrared spectra of $Cu_{50}Co_{50}$ viz. (1) At the working potential of 0.2 V, close to the onset potential of the LSV curve, the upward absorption bands at 1392 and 1354 $cm^{-1}$ were ascribed respectively to N-O symmetric and asymmetric stretching vibration of $NO_3^-$[38], indicating consumption of $NO_3^-$ species in the thin layer; (2) at the same time, the downward band at 1236 $cm^{-1}$ appeared and was attributed to N-O antisymmetric stretching vibration of $NO_2^-$[39], indicating $NO_2^-$ formation from $NO_3^-$ reduction; (3) with potential negatively moving to 0.1 V, another intermediate observed around 1110 $cm^{-1}$ was ascribed to -N-O- stretching vibration of hydroxylamine ($NH_2OH$)[39,40], which was a key intermediate for $NH_3$ formation; (4) The upward band around 1638 $cm^{-1}$ was attributed to water electrolysis responsible of hydrogen generation involved in the hydrodeoxidation of $NO_3^-$ in the solution of thin-layer[41].

The FTIR spectra collected on Cu (Fig. 4b) were very similar to the $Cu_{50}Co_{50}$ catalyst's ones, indicating that the $NO_3^-$RR behaviors were similar for both catalysts. However, the $NO_3^-$ consumption on Cu at a

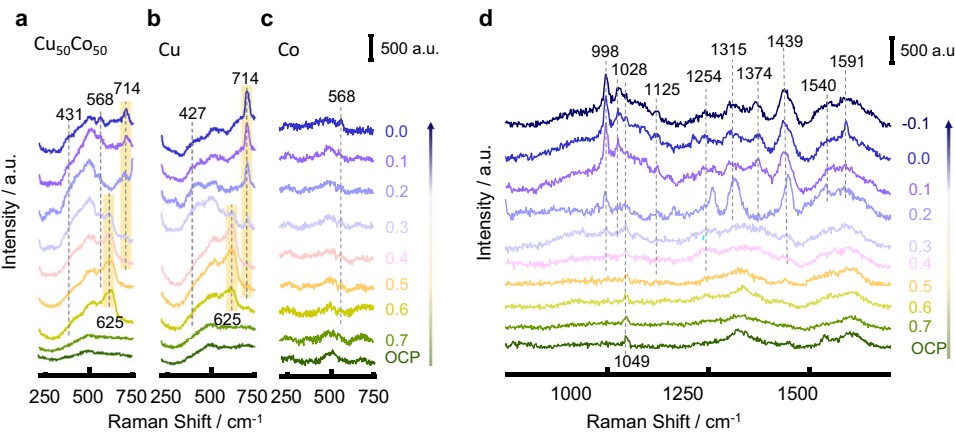

**Fig. 5 | Electrochemical SHINERS spectra of $NO_3^-$RR.** SHINERS spectra between 230–750 $cm^{-1}$ on $Cu_{50}Co_{50}$ (**a**), Cu (**b**), and Co (**c**). SHINERS spectra between 750–1700 $cm^{-1}$ on $Cu_{50}Co_{50}$ (**d**) in 100 mM $KNO_3$ + 10 mM KOH during cathodic polarization from 0.7 to −0.1 V.

potential of 0.1 V was 100 mV more negative than the one obtained on $Cu_{50}Co_{50}$, indicating a better kinetic with the latter one. In addition, with the negative shift of the working potential, the band intensity of $NO_2^-$ relative to the sum intensity of $NO_2^-$ and $NH_2OH$ production (i.e., $\frac{I_{NO_2^-}}{I_{NH_2OH} + I_{NO_2^-}}$) on $Cu_{50}Co_{50}$ dropped down sharply, while the $\frac{I_{NO_2^-}}{I_{NH_2OH} + I_{NO_2^-}}$ ratio was almost independent of the potential on Cu (Fig. 4d). The two phenomena, i.e., the appearance of $NH_2OH$ after the formation of $NO_2^-$ and the increase of $NH_2OH$ at the expense of the consumption of $NO_2^-$, suggested that $*NH_2OH$ was obtained by the deep hydrogenation of $*NO_2$ [42]. The $\frac{I_{NO_2^-}}{I_{NH_2OH} + I_{NO_2^-}}$ ratio on $Cu_{50}Co_{50}$ was much lower than the one on Cu catalyst, demonstrating that alloying Cu with Co could deeply enhance the hydrogenation of $*NO_2$ to the final product $NH_3$ on Cu/Co alloy catalysts. Thereby we speculated that Co sites were responsible for the variation of the $\frac{I_{NO_2^-}}{I_{NH_2OH} + I_{NO_2^-}}$ ratio, owing to their excellent protons' adsorption ($*H$) capacity, as it will be demonstrated in the next section, promoting the hydrodeoxidation of $*NO_2$ to $*NH_2OH$ according to Eqs. (3) and (4):[43]

$$*NO_2 + 2*H \rightarrow *NO + H_2O \tag{3}$$

$$*NO + 3*H \rightarrow *NH_2OH \tag{4}$$

On pure Co (Fig. 4c), no spectra bands of $NO_3^-$RR were detected until the working potential negatively shifted to 0 V. The absence of $NO_2^-$ on the Co catalyst suggested a low $NO_2^-$ accumulation in the thin layer solution, corroborating very well first the results obtained from the K–L equation where the Co catalyst was more inclined to perform continuous hydrogenation of $NO_3^-$ to $NH_3$ via an 8-electrons transfer and second the lower Tafel slope in the Peak S2 region data.

In comparison with thin-layer in situ FTIR, attenuated total reflection in situ FTIR analysis (ATR-FTIR) is more sensitive to the signal of adsorbed species on catalysts' surfaces[44]. Two weak vibration bands of adsorbed NO in two different adsorption modes ("bridge" and "on top") were detected at 1557 and 1639 $cm^{-1}$ on the $Cu_{50}Co_{50}$ catalyst (Supplementary Fig. S29a)[38,45]. Interestingly, a downward band center at 2109 $cm^{-1}$ at 0.5 V was observed in $NO_3^-$-free electrolyte on $Cu_{50}Co_{50}$ (Fig. 4e), and the band center was negatively shifted to 2085 $cm^{-1}$ at −0.4 V, yielding a Stark turn rate of 26.7 $cm^{-1}$ $V^{-1}$. These IR features could be attributed to adsorbed $*H$ on the $Cu_{50}Co_{50}$ surface, and indicated also the enhancement of $*H$ adsorbed on the $Cu_{50}Co_{50}$ surface at a higher η, which is consistent with the reported studies[46,47]. The fact that $*H$ was observed on $Cu_{50}Co_{50}$ and Co (around 2110 $cm^{-1}$) (Fig. 4e, g and Supplementary Fig. S30c) while being absent on Cu

(Fig. 4f and Supplementary Fig. S30b), indicated that $*H$ on $Cu_{50}Co_{50}$ was mainly attributed to water dissociation on Co sites. When the potential negatively shifted, the intensity of Co-H gradually raised, demonstrating that more $*H$ were generated which in turn enhanced the hydrodeoxidation reaction and gave a lower $\frac{I_{NO_2^-}}{I_{NH_2OH} + I_{NO_2^-}}$ ratio value.

The wavenumber attributed to $*H$ on Co was slightly higher than the one on $Cu_{50}Co_{50}$, indicating a stronger affinity of $*H$ for pure Co catalyst. In the presence of $NO_3^-$, Co-H was still present in the spectra of pure Co (Supplementary Fig. S30c), while it was vanished in the spectra of $Cu_{50}Co_{50}$ (Supplementary Fig. S30a). This could be explained by the fact that $*H$ could adsorb quickly on Co sites inhibiting the adsorption of $NO_x$, then the adsorbed hydrogen can react with the equivalent amount of $*NO_x$ on nearby Cu sites giving a high reaction kinetics. In the case of monometallic Co, the active sites are majorly occupied by $*H$ species preventing the adsorption of $NO_x$ onto the catalyst's surface, leading to weak activity. The presence of active hydrogen (H*) in the reaction process was also verified by electron paramagnetic resonance (EPR) analysis on $Cu_{50}Co_{50}$ catalyst (Supplementary Fig. S31) using 5,5-dimethyl-1-pyrroline N-oxide (DMPO) as a spin trap[48]. The intensity of the EPR signal of the DMPO-H adduct on the $Cu_{50}Co_{50}$ catalyst decreased when nitrate was added into the electrolyte, indicating the consumption of active hydrogen during $NO_3^-$RR. These results were consistent with the ATR-FTIR analysis. Based on the coupling of thin-layer in situ FTIR and ATR-FTIR analysis, therefore, we proposed the following pathway for the $NO_3^-$RR on $Cu_{50}Co_{50}$: $NO_3^- \rightarrow *NO_3 \rightarrow *NO_2 \rightarrow *NH_2OH \rightarrow *NH_3$, where $*H$ on Co sites can promote the deep hydrodeoxidation of $NO_2^-$ to $NH_2OH$.

## SHINERS analysis of $NO_3^-$RR

The reaction intermediates provided by electrochemical in situ FTIR spectra were still insufficient to figure out the overall roadmap of $NO_3^-$RR to $NH_3$, in this aim, SHINERS spectra of the catalysts were collected to probe the catalysts' surface during the reaction. Supplementary Table S4 compiled the Raman scattering peaks observed during $NO_3^-$ reduction on $Cu_{50}Co_{50}$, Cu, and Co, which were not detected on Au@SiO$_2$ (Supplementary Fig. S32). The wavenumber below 750 $cm^{-1}$ corresponds mainly to the chemical properties of the catalyst's surface[43]. SHINERS spectra of $Cu_{50}Co_{50}$, Cu, and Co, in this section, were summarized and shown in Fig. 5. At 0.6 V, a characteristic band at 625 $cm^{-1}$ associated with $Cu_2O$[49] was observed on Cu and $Cu_{50}Co_{50}$ (Fig. 5a, b), indicating partial oxidation of the catalysts surface due to air exposition before $NO_3^-$RR; these results were consistent with the XPS data. As the working potential decreased to 0.3 V, the band intensity of $Cu_2O$ gradually shrank, and a peak at 714 $cm^{-1}$ emerged in replacement and was associated with the bending mode of

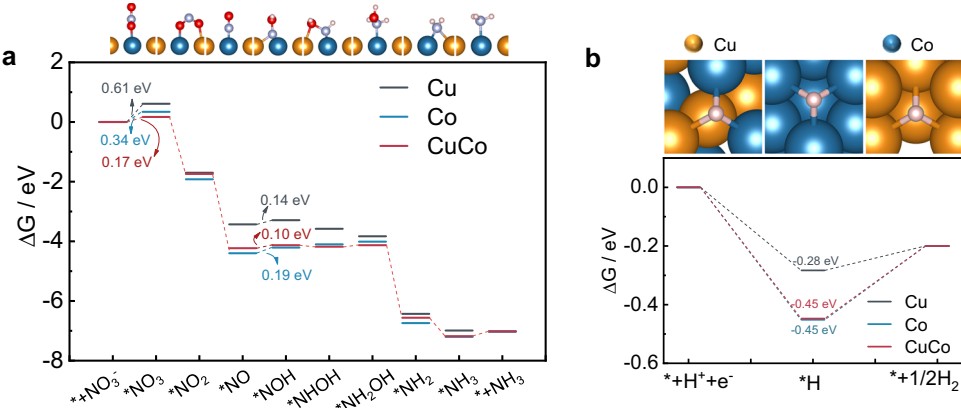

**Fig. 6 | DFT calculations of NO$_3^-$RR and HER on Cu(111), Co(111), and CuCo(111).** Reaction-free energies for different intermediates of NO$_3^-$RR (**a**) and HER (**b**) at −0.2 V *vs.* RHE on CuCo(111), pure Cu(111), and pure Co(111) surface, respectively.

free Cu-OH$_{ad}$[50,51], indicating a gradual reduction of Cu before NO$_3^-$RR started to occur at 0.2 V. Besides, the band at 431 cm$^{-1}$ can be assigned to Cu-O$_x$ due to the adsorption of oxynitride on Cu surface[51], and the band's intensity increased as the potential moved negatively. For Co (Fig. 5c), a band at 568 cm$^{-1}$ was also observed, associated with the formation of Co-O$_x$ caused by the same oxynitride species adsorbed on the catalyst's surface[52].

In addition, the signals of NO$_3^-$ and its reduced intermediates adsorbed on the catalysts' surface were observed in the spectra between 750 and 1700 cm$^{-1}$. As the potential decreased from 0.7 to −0.1 V, several peaks appeared in sequences on Cu$_{50}$Co$_{50}$ (Fig. 5d) viz. (1) At open circuit potential (OCP) and 0.7 V, the only obvious peak viewed at around 1049 cm$^{-1}$ was attributed to the symmetric NO$_3^-$ stretching from solution NO$_3^-$ species[53]. (2) When the potential was decreased to 0.4 V, the NO$_3^-$ species in the solution started to adsorb on the surface of Cu$_{50}$Co$_{50}$ since four peaks appeared in three different forms as NO stretching vibration from the unidentate nitrate near 998 cm$^{-1}$, symmetric and antisymmetric stretching vibration of the NO$_2$ group from the chelated nitro configuration around 1125 and 1254 cm$^{-1}$[54], and the N=O stretching vibration of the bridged nitro group closed to 1439 cm$^{-1}$, respectively[55]. (3) As the working potential shifted negatively further to 0.2 V, the symmetric NO$_3^-$ stretch of *NO$_3^-$ appeared around 1028 cm$^{-1}$ [56,57] and symmetric bending vibrations of the HNH near 1315 and 1374 cm$^{-1}$ were apparent;[55,57] N=O stretch of HNO[56] was visible around 1540 cm$^{-1}$. (4) When the working potential negatively shifted further to 0.1 V, the HNO peak near 1540 cm$^{-1}$ disappeared quickly; meanwhile, the antisymmetric bending vibration of the HNH of NH$_3$ at 1591 cm$^{-1}$ came out[58], which indicated an efficient formation of NH$_3$ from HNO hydrodeoxidation.

The SHINERS spectra on Cu and Co catalysts (Supplementary Fig. S33a, b) were similar to the one of Cu$_{50}$Co$_{50}$, but a new peak close to 800 cm$^{-1}$ related to the bending vibration of NO$_2^-$ group was only observed on Cu (Supplementary Fig. S33a)[26]. This could be due to the accumulation of NO$_2^-$ species near by the Cu surface due to its poor ability for deep NO$_2^-$ hydrodeoxidation properties. The intensity of $\nu_{s(NO_3^-)}$ from NO$_3^-$ species in solution on Co at low overpotential was similar to the one on Cu$_{50}$Co$_{50}$ and Cu, however, the band's intensities of all the intermediates formed on Co were very weak (Supplementary Fig. S33b), meaning that a low amount of *NO$_3^-$ species and its derivates were adsorbed on the Co surface. It can be rationally speculated that the affinity of *H species to the Co surface was such strong that the active sites on the Co catalyst were majorly occupied by *H species leading to low coverage of *NO$_3^-$ species, bringing a FE$_{NH_3}$ close to 85% (Fig. 3b) at the expense of a low current density of NO$_3^-$RR of 25.4 mA cm$^{-2}$ (Supplementary Fig. S24a) at 0 V. Similarly, *H species affinity for Cu surface was so weak that most active sites were occupied by *NO$_3^-$ species, leading to a relatively high current density up to

104.8 mA cm$^{-2}$ (Supplementary Fig. S24a) and yet an unsatisfactory FE$_{NH_3}$ of 32% (Fig. 3b). Therefore only if a balanced coverage of *H and *NO$_3^-$ species is achieved, a satisfactory FE$_{NH_3}$ and $j$ can be simultaneously obtained, which is the case with Cu$_{50}$Co$_{50}$ where FE$_{NH_3}$ and $j$ were 88% 394.6 mV respectively.

With combined electrochemical in situ FTIR and SHINERS spectroscopic analysis, the NO$_3^-$RR pathway on Cu, Co and CuCo was proposed as a series of deoxygenation reactions according to the following path, NO$_3^-$ → *NO$_3^-$ → *NO$_2^-$ → *NO, accompanied by a series of hydrogenation reactions: *NO → *NOH → *NH$_2$OH → * NH$_3$ → NH$_3$.

## DFT calculations

Based on all the aforementioned observations and to shed light on the NO$_3^-$RR mechanism on Cu, Co, and CuCo at the atomic level, the density functional theory (DFT) calculations were conducted and the Gibbs free energies (ΔG) of *NO$_3$ species and their derivates on Cu(111), Co(111) and CuCo(111) surface were presented in Fig. 6a.

In terms of thermodynamics, the rate-determining step (RDS) on Cu(111), Co(111) and CuCo(111) was the initial *NO$_3$ adsorption step and lower energy was required on CuCo(111) (0.17 eV), compared to Cu(111) (0.61 eV) and Co(111) (0.34 eV), indicating stronger adsorption of NO$_3^-$ species on CuCo(111)[15,59]. Interestingly, the ΔG of the initial *NO$_3$ species on Co(111) was lower than the one obtained on Cu(111), which should lead to better NO$_3^-$RR performances. However, this result contradicted the fact that $\eta_{@10mAcm^{-2}}$ of NO$_3^-$RR on Cu was 503 mV more positive than on Co ($\eta_{@10mAcm^{-2}}$ of 690 mV). This inconformity could be explained by a strong competition of *H species on the catalyst sites' surface compared to nitrogenous species, as proved by in situ FTIR and SHINERS analysis. Indeed, Co was capable of promoting hydrogenation during the NO$_3^-$RR by transferring in situ *H species[13], because of its excellent adsorption for *H species[60]. However, the excessive adsorption of *H can reduce the coverage of *NO$_3$ species and their intermediates, thereby weakening the electrochemical activity toward NO$_3^-$RR. Compared with the ΔG of *NO$_3$ and *H on Co(111) (Fig. 6b and Supplementary Table S11), the adsorption for *H was stronger, but the adsorption for NO$_3^-$ was poor. With the incorporation of the Cu atom, the electronic structure changed, enhancing the adsorption of NO$_3^-$ species on CuCo(111) and furthermore, the partial replacement of Co sites with Cu sites also reduced the coverage of *H species.

Based on the comprehension of the results discussed above, the redistribution of electrons in Cu$_{50}$Co$_{50}$ facilitated the electrons transfer rate and uplifted the catalytic kinetic of the NO$_3^-$RR (Supplementary Figs. S6, S7). The fair coverage of *H and *NO$_3$ species were important to obtain simultaneously a high FE$_{NH_3}$ and $j_{NH_3}$. Co-adjustment of Cu and Co sites on Cu$_{50}$Co$_{50}$ can balance the adsorption energy between *H and *NO$_3$ species, not only by lowering the

energy barrier of $NO_3^-$ reduction but also by improving the hydrogenation capability with the enhanced *H adsorption compared with pure Cu, and at the same time avoiding the excessive *H occupation of the active sites compared to pure Co.

To conclude, we presented a novel Cu-based catalyst able to mimic the Cu core center of the Cu-NIR for the electrocatalytic $NO_3^-$ reduction. The addition of Co to Cu formed a micro-pine structure with nanosheets and enhanced dramatically the proton availability over the surface of the catalyst from water electrolysis, resulting in an almost full Faraday efficiency of $NH_3$ production at an ampere-level current density of $1035\,mA\,cm^{-2}$ at $-0.2\,V$. This process exhibited a high activity and could be proposed as a sustainable and eco-friendly complementary route of $NH_3$ production. In batch conditions, the catalyst was able to achieve a $FE_{NH_3}$ of 96% and a nitrate's removal of 99.5%, from an initial concentration of 6,200 ppm, reaching a final concentration of 31 ppm after 10 h of reaction, lower than the limitations fixed by the World Health Organization for drinking water. A mechanism was established by combining in situ FTIR and SHINERS spectroscopic investigations and DFT calculations. The synergy between Cu and Co can reduce the high energy barrier of the rate-determining step of the initial *$NO_3$ adsorption step on Cu, due to a regulation of the electronic structure. The $\Delta G$ of the hydrogenation of the intermediate species, *$NO_x$, could also be reduced due to the facile adsorption of *H species on Co(111) compared to Cu(111). We proposed that a rational control of the *H adsorption over the surface of the bimetallic catalyst is the key to managing further adsorption of intermediates from $NO_3^-$RR to achieve excellent performances. This discovery can provide an additional dimension to research into surface adsorption-modulated bimetallic catalysts for highly reactive hydro-deoxidation reactions and can provide a novel strategy for the development of multi-component heterogeneous catalysts for efficient $NH_3$ production and wastewater treatment.

## Methods

### Preparation of $Cu_xCo_y$, Cu, and Co catalysts

The catalysts were prepared via an electrodeposition process under the current density of $50\,mA\,cm^{-2}$ for 300 s in a two-electrodes system where a platinum plate was used as the counter electrode and the Ni foam as the working electrode. Ni foams were pretreated with acetone and ethanol, and then repeatedly rinsed with ultrapure water and dried under a heating lamp. The deposition electrolyte (50 mL) comprised of 0.015 M trisodium citrate pentahydrate solutions and 50 mM $CuSO_4 + CoSO_4$[61,62]. The $Cu_xCo_y$ catalysts with Cu:Co ratios of 65:35, 50:50, and 15:85, denoted as $Cu_{65}Co_{35}$, $Cu_{50}Co_{50}$, and $Cu_{15}Co_{85}$, were obtained in the deposition solutions with the ratio of $CuSO_4$ to $CoSO_4$ as 60:40, 45:55, and 10:90, respectively. The Cu and Co catalysts were prepared. The preparation for pure Cu and Co catalysts followed the same steps as the $Cu_xCo_y$ synthesis, except that only $CuSO_4$ or $CoSO_4$ solution was present in the electroplating solution. The catalysts were finally rinsed with ultrapure water and dried under the protection of Ar.

### Material characterization

Morphology and elemental composition were characterized using a scanning electron microscope (SEM, ZEISS Sigma) with an energy dispersive X-ray spectrometer at an operating voltage of 15 kV. The lattice arrangement was observed using a high-resolution transmission electron microscope (HRTEM, FEI-Tecnai G2 F20) at an accelerated voltage of 200 kV. The chemical composition was analyzed by an inductively coupled plasma emission spectrometer (ICP-OES, Thermo Fisher iCap 7000). X-ray diffractometer (XRD, SmartLab-SE) with Cu-Kα X-ray source was used for crystal material structure analysis. Al Ka X-ray excited Thermo Fisher Scientific Nexsa X-ray Photoelectron spectrometer (XPS, Nexsa) was used for chemical state analysis. All XPS spectra were corrected with a C 1 s spectral line of 284.8 eV. X-ray absorption fine structure (XAFS) spectra at Cu K-edge and Co K-edge

were obtained on the 1W1B beamline of Beijing Synchrotron Radiation Facility (BSRF) operated at 2.5 GeV and 250 mA. Standard data processing, including energy calibration and spectral normalization of the raw spectra was performed using Athena software.

### Electrochemical test

The electrochemical measurements were performed using a three-electrodes system connected to the CHI 760E workstation (Chenhua, Shanghai) in a homemade H-type cell (separated by Nafion 117 membrane; with magnetic stirring of 1000 rpm). The Nafion 117 was pre-processed according to the reported procedures. The $Cu_xCo_y$/Ni foams ($0.5\,cm^2 \times 0.5\,cm^2$) were used as working electrodes, and platinum plate and Hg/HgO electrode (filled with 1 M KOH solution) were used as counter and reference electrodes, respectively. Before the testing, all the catalysts were electro-reduced at $-0.2\,V$ vs. RHE for 600 s in 1 M KOH solution to eliminate surface oxidation. 1 M KOH aqueous solution containing different $KNO_3^-$ concentrations (5, 10, 50, and 100 mM) were used as an electrolyte (30 ml). The electrolyte was bubbled with Ar to remove $O_2$ and $N_2$ for 10 min before the experiment. The electrochemical linear voltammetry (LSV) curves were obtained in a single cell. The current density was normalized to the geometric electrode area ($0.25\,cm^2$) unless otherwise specified. The cyclic voltammetry curves in electrochemical double-layer capacitance ($C_{dl}$) determination were measured in a potential window where no Faradaic process occurred in an electrolyte of 1 M KOH at different scanning rates of 20, 40, 60, 80, and $100\,mV\,s^{-1}$. All the potentials were converted to the RHE reference scale by $E_{(V\ vs.\ RHE)} = E_{(V\ vs.\ Hg/HgO)} + 0.0591 \times pH + 0.098$. Note that $NH_3$ volatilization in the 1 M KOH electrolytes (pH 13.6) is negligible during the 1-h electrolysis.

### Kinetic evaluation

The electrochemical kinetic analysis of $NO_3^-$RR was performed based on the Koutecký–Levich (K–L) equation, as shown in Eq. (5):

$$\frac{1}{i_m} = \frac{1}{i_K} + \frac{1}{0.2nFD^{2/3}\nu^{-1/6}C\omega^{1/2}} \tag{5}$$

Where $i_m$ is the test current; $i_K$ is the kinetic current of $NO_3^-$ reduction; $n$ is the number of electrons transferred in the reaction; $F$ is the Faraday constant, $96485\,C\,mol^{-1}$; $D$ represents the effective diffusion coefficient of $0.1\,mol\,L^{-1}$ $NO_3^-$ at 25 °C, $1.4 \times 10^{-5}\,cm^2\,s^{-1}$; $\upsilon$ represents the kinematic viscosity of water at 25 °C, $1 \times 10^{-6}\,m^2\,s^{-1}$; $C$ is $NO_3^-$ concentration, $mmol\,L^{-1}$; $\omega$ is the electrode speed, rpm.

The working electrode was prepared as follows: (1) 5 mg of catalyst powder dropped down from $Cu_xCo_y$/Ni foam and was dispersed in the solution of $600\,\mu L$ isopropanol $+ 380\,\mu L$ ultrapure water $+ 20\,\mu L$ 5% Nafion solution, and then sonicated for at least 1 h to get a homogeneous ink; (2) $10\,\mu L$ ink was drop-casted onto the rotating disk electrode (Ø, $0.196\,cm2$) with the loading of $0.255\,mg_{catalyst}\,cm^{-2}$ for the further LSV analysis at different speeds (100, 225, 400, and 625 rpm) with a scan rate of $10\,mV\,s^{-1}$. A platinum electrode and an Hg/HgO electrode were used as counter electrode and reference electrode, respectively. An aqueous solution of $1\,mol\,L^{-1}$ KOH containing $100\,mmol\,L^{-1}$ $KNO_3$ was used as the electrolyte. Ar was used to purge the dissolved $O_2$ and $N_2$ from the electrolyte.

### Product detection and efficiency calculation

The $NH_3$ concentration was quantified by the Nessler Reagent method[28]. The electrolytes sampled after electrolysis for 1 h were first neutralized with $0.5\,M\,H_2SO_4$ and then mixed with $6.25\,mL$ ultrapure water and $0.25\,mL$ of Nessler reagent for the chromogenic reaction. The absorbance of the mixed solution was measured at a wavelength of 420 nm after keeping it at room temperature for 30 min. The quantitation of $NH_3$ was performed by the standard curves, which was built using standard $NH_4Cl$ solutions in 1 M KOH. The $^1H$ NMR

(500 MHz) determination[31] was also carried out to quantify the $^{14}NH_4^+$ and $^{15}NH_4^+$ after electrolysis at −0.2 V *vs.* RHE for 1 h. The electrolytes were mixed with 0.4 M $H_2SO_4$ at a ratio of 500:125 to ensure adequate protonation of $NH_3$. Then, 125 µl of the diluted electrolytes or standard solution were mixed with 125 µl of 10 mM maleic acid in DMSO-D6, 50 µl of 4 M $H_2SO_4$, and 300 µl of $H_2O$.

The $NO_3^-$ and $NO_2^-$ in the solution were quantitatively determined by ion chromatography (IC). The possible gas products of $NO_3^-RR$, such as $H_2$, $N_2$, NO, $NO_2$, $N_2O$, and $NH_3(g)$ were analysed using a gas chromatography (GC) and online electrochemical mass spectrometry (OEMS)[30].

The Faradaic efficiency was calculated according to the following equation:

$$FE = \frac{ncV_{catholyte}F}{Q} \times 100\% \tag{6}$$

where $c$ represents the concentration of the product, mol $cm^{-3}$; $V_{catholyte}$ is the volume of catholyte, mL; $Q$ is the total amount of charge consumed, C.

The yield rate of $NH_3$ was calculated according to the following equations:

$$Yield_{NH_3} = \frac{cV_{catholyte}}{St} \tag{7}$$

$$Yield_{mass-NH_3} = \frac{cV_{catholyte}}{mt} \tag{8}$$

where $S$ is the area of the geometrical cathode, $cm^{-2}$; $m$ is the mass of the catalyst on the cathode; $t$ is the time of the electrolysis.

### Electrochemical in situ FTIR reflection analysis

Electrochemical in situ FTIR reflection spectroscopy[44]. Electrochemical thin-layer in situ FTIR spectroscopy measurements were performed on a Nicolet Nexus 8700 FTIR spectrometer equipped with a liquid $N_2$-cooled system and MCT-A detector. The glassy carbon electrode loading with catalysts was used as the working electrode, which was pressed vertically on the $CaF_2$ window plate to form a thin liquid layer with a thickness of about 10 µm. A platinum foil and an Hg/HgO electrode (filled with 1 M KOH solution) were used as the counter electrode and reference electrode, respectively. The incoming infrared beam was approximately aligned with the normal electrode surface. Unless otherwise noted, the sample spectra were averaged from 200 interference spectra with a resolution of 8 $cm^{-1}$. Reference spectrum ($R_{Ref}$) were collected at 0.4 V, and sample spectra ($R_S$) were collected in the potential region from 0.4 V to −0.2 V and stepped by 100 mV. The spectra were reported as $\Delta R/R = (R_S − R_{Ref})/ R_{Ref}$.

Attenuated Total Reflection in situ FTIR reflection spectroscopy[36]. The gold-plated Si prism with catalysts were assembled into a home-made spectral-electrochemical cell, which contained a carbon sheet as a counter electrode and Hg/HgO electrode as a reference electrode. It was then fixed in a homemade optics system built in the chamber of a Nicolet Nexus 8700 FTIR spectrometer for electrochemical ATR-FTIR measurements at an incidence angle of *ca.* 65°. The ATR-FTIR spectra were reported in the same way of thin-layer in situ FTIR, except that $R_{Ref}$ was taken at 0.5 V and $R_S$ were collected in the potential region from 0.5 to −0.4 V.

### SHINERS analysis

SHINERS spectra were recorded in a custom-made in situ Raman spectroelectrolysis cell using an XplorA confocal microprobe Raman spectrometer (HORIBA Jobin Yvon)[63]. The excitation wavelength of the laser was 637.8 nm and came from a He-Ne laser with a power of about 6 mW. The electrochemically polished gold electrode (diameter 3 mm) was modified by 10 µL catalyst ink with 10 µL homemade shell-isolated gold nanoparticle, which was provided by Prof. Jian-Feng Li at Xiamen University, China, and applied as the working electrode. Hg/HgO electrode was used as the reference electrode, and platinum wires was used as the counter electrode. A long-focus objective (8 mm) of A × 50 magnification was used. A Si wafer (520.6 $cm^{-1}$) was used to calibrate the Raman frequency before the experiment. The SHINERS spectra were obtained using the cumulative results of four tests for 30 s each.

### EPR Experiments

5,5-dimethyl-1-pyrroline *N*-oxide (DMPO) was used to capture the instable hydrogen radical to form the DMPO-H adduct to generate EPR spectra[64]. In the experiments, 5 ml electrolyte was mixed with 10 µL DMPO and was deoxygenated by bubbling Ar. The potentiostatic electrolysis was carried out for 5 min in the H-type cell under the protection of Ar. EPR measurement was performed by Bruker EMX-10/12 spectrometer operating at a frequency near 9.5 GHz, sweep width of 200 G and power of 20 mW.

### DFT calculations

All DFT calculations in this work were carried out with the Vienna Ab initio Simulation Package (VASP)[65]. And the projector augmented-wave (PAW) pseudopotential was selected to deal with the core-valence interaction[66]. The generalized gradient approximation (GGA) of Perdew–Burke–Ernzerhof (PBE) was used to account for the exchange and correlation of electronics and the cut-off energy of plane-wave was 600 eV[67]. The energy convergence criterion was within $10^{-5}$ eV and the Hellmann–Feynman force was smaller than 0.01 eV $Å^{-1}$ on each atom. The converged unit cell models of Cu (3.64 × 3.64 × 3.64 $Å^3$), Co (3.52 × 3.52 × 3.52 $Å^3$), and CuCo (3.78 × 3.49 × 3.49 $Å^3$) were used in DFT calculations, respectively. The dimension of a 2 × 2 supercell of Cu (111) (8.91 × 10.28 $Å^2$), a 2 × 2 supercell of Co (111) (8.62 × 9.95 $Å^2$) and a 2 × 2 supercell of CuCo (111) (9.03 × 9.87 $Å^2$) were used, respectively. These supercells were constructed and contained three layers and a sufficient vacuum layer of 15 Å thicknesses. For the structural optimization, the bottom two layers were fixed and the top layer was fully relaxed[68]. For unit cell geometry optimization, an 8 × 8 × 8 k-point analysis was used. A grid of 3 × 3 × 1 k-point mesh was used for these supercell calculations[15]. The calculations of all molecules and intermediate species on Co(111) and CuCo(111) were performed with spin polarization[69]. Dipole corrections in the z direction were included in all computations to minimize inaccuracies in the total energy because of simulated slab interactions. The spin polarization was not taken into account in the calculations of intermediate species on Cu(111) due to the spin polarization did not affect the Cu (111) calculations.

## Data availability

The raw data of the figures in the main manuscript are available in figshare with the identifier(s) https://doi.org/10.6084/m9.figshare. 21671075. All other data needed to evaluate the conclusions in the paper are present in the paper and the Supplementary Information or can be obtained from the corresponding authors on reasonable request.

## Code availability

The code used in this work can be obtained from the corresponding authors on reasonable request.

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

## Acknowledgements

This research was financially supported by the National Natural Science Foundation of China (NSFC) (Nos. 22002131, Y.-Y.L.) and China Post-doctoral Science Foundation (Grant No 2020M671963, Y.-Y.L.). X.-Y.H. thanks R.O. from Seoul National University for the fruitful discussions. We are thankful to the Beijing Synchrotron Radiation Facility (1W1B, BSRF) for their help with the characterizations.

## Author contributions

Y.-Y.L., J.-Y.F., & S.-G.S. contributed to the conception of the study. J.-Y.F. performed the experiments. S.-N.H. provided assistance with the HRTEM analysis. K.-M.Z. helped with the SHINERS analysis. G.L. helped with the electrochemical in situ FTIR analysis. Q.-Z.Z. performed the DFT calculations analysis. J.-Y.F. & Y.-Y.L. performed the data analyses and wrote the manuscript. X.-Y.H., O.A., & S.-G.S. contributed significantly to the analysis and manuscript preparation. The project was supervised by Y.-Y.L. and S.-G.S. All authors helped perform the analysis with constructive discussions.

## Competing interests

The authors declare no competing interests.
