## [Peer Review File · Nature Communications]

REVIEWER COMMENTS

Reviewer #1 (Remarks to the Author):

The manuscript by Sun et. al., reported the enzyme-inspired design of Cu₅₀Co₅₀ nanosheet as an efficient electrocatalyst for the nitrate reduction reaction. The authors performed a series of in-situ FTIR spectroscopy, as well as DFT calculation, to investigate the synergy of Cu and Co in promoting the nitrate reduction. Considering the deep mechanism study and good electrocatalytic performance, I suggest to accept this manuscript after pending minor revisions. Some specified comments are listed as follows:

1. In "Abstract", the authors stated that "a $100 \pm 1\%$ Faradaic efficiency at an ampere-level current density of 1035 mA cm^{-2} for NH_3 production at -0.2 V vs. RHE ." However, as shown in Figure 2a, the current density of Cu₅₀Co₅₀ nanosheet at -0.2 V vs. RHE is $\sim 300 \text{ mA cm}^{-2}$. Please explain why?
2. In this work, the authors emphasized the strategy of enzyme-inspired design to boost the activity of nitrate reduction. However, in "Abstract", the authors did not mention how to achieve enzyme-inspired design.
3. More gaseous products (i.e., N_2O , NO_2 , NH_3) should be quantified. Especially, part of the generated NH_3 can probably be stripped out from electrolyte in the alkaline solution (1 M KOH).
4. The quantification methods should be at least two types. The authors should refer to the recent advances in the electrochemical reduction of N_2 that introduce how to quantify ammonia. Also, the ammonia yield in electrolyte free of nitrate should also be tested.
5. More experimental data are expected to support the durability of the catalysts. How about the variation in the performance of ammonia yield and Faradaic efficiency during the consecutive recycling test? Were the catalysts stable during the nitrate reduction? More characterizations and comparisons are needed for the catalysts before and after the measurements.
6. Electrochemical nitrate reduction in an alkaline electrolyte involves eight electrons transfer coupled with the generation of nine OH^- , so what was the H^+ source of the final NH_3 product? The pH change during the nitrate reduction in alkaline media should be monitored.
7. Considering that the pH of waste water is mostly ~ 7 , the authors should test the performance of different electrocatalysts in neutral media.
8. After alloying with Co, the lattice spacing of Cu (111) plane is contracted. In general, the lattice constrain of metal sites tunes the adsorption strength towards reaction intermediates and thus the reaction activity. Did the lattice constrain of Cu also impact the activity?

Reviewer #2 (Remarks to the Author):

The authors present a study of nitrate reduction to ammonia on CuCo catalysts, finding improved overpotentials relative to Cu and Co references. This is an important topic, though the study could use significant clarification and refinement before it is suitable for publication.

I don't particularly understand what is bio-inspired or enzyme-like about these catalysts. A hallmark of enzyme catalysis is the specificity; this would not seem to be the case here. I felt a rather weak case was made when trying to bridge Cu-containing reductases with metallic catalysts. The paper may be improved by eliminating this biological comparison, if no stronger connections to enzyme form/function can be made.

Relatedly, the nature of this catalyst is still unclear to me. The authors describe these as CuCo sheets, though the images in Figure 1 and Figure S2 seem to suggest otherwise. If they are sheets, how does that relate to the enzyme-like behavior? A more clear and consistent description of the catalyst would help.

The authors compare NH₃ production rates on a mass basis with the Haber-Bosch catalyst. I'm not sure this is an appropriate comparison, given the highly disparate costs of Cu and Fe catalysts, as well as the major differences between a lab-scale electrochemical setup and the industrial scale Haber-Bosch process.

What is the role of the Ni foam? Would this process work if simply depositing Co on a Cu foam?

There are several items related to the calculations that must be clarified for reproducibility:

(i) The lattice constants used in calculations should be provided. What is the unit cell size? (This should not need to be inferred from SI figures.) Which atoms were relaxed or fixed in the slabs? What k-point sampling was used? Were calculations performed with spin polarization? Was a dipole correction included for the slab calculations?

(ii) How are free energies calculated, and what are the reference states? Relatedly, how was the adsorption energy of the negatively charged species modeled?

(iii) All calculated energies for intermediates should be provided in SI tables.

(iv) How does NO₃ dissociation occur? If it is assumed to occur by direct N-O bond scission, there is an energy barrier associated with this that may be larger than other calculated free energy barriers. The implications of activation energy barriers should be discussed, if not explicitly calculated.

(v) Why was an fcc model used for Co, which has an hcp structure? I was expecting to see Co(0001) used.

(vi) How was the bimetallic constructed, including determination of an appropriate lattice constant? Is there experimental evidence for alternating rows of Co and Cu? The authors may be missing important features about potential active sites if restricting things to these highly ordered sites. Limitations should be briefly discussed.

(vii) The authors do not specify an electrochemical model, which is typically required to get accurate insights into reaction energetics and identify quantities such as the rate determining step in electrochemical studies. If the authors intended to provide energetics at 0.0 V(RHE) and have used H₂ gas as a reference, they may have serendipitously ended up with energetics and conclusions that would be obtained with, e.g., Norskov's computational hydrogen electrode. Still, such a model should be applied and the potential used to calculate reaction energetics should be specified with all corresponding figures. (I might suggest presenting results at -0.2 V(RHE), if that is indeed the most interesting potential experimentally.)

(viii) On Page 18, the authors discuss a potential redistribution of electrons in Cu₅₀Co₅₀ and subsequent effects on reaction kinetics. The rationale for this statement is unclear, as I don't see computational evidence for such a redistribution of electrons nor any calculated kinetic quantities.

Minor comments:

In Figures 6 and S22, can different colors be used for Cu and Co? It is difficult to distinguish the two.

The writing quality could be improved, particularly in the methods section. There are places where the meaning could be confused: the authors mix the use of "absorption" and "adsorption", which have different meanings. There is a typo even in the title of the article: "Haber-Bosch"

It is unusual that the methods section was written without any external references; there are typically references provided to previously-used techniques, computational codes, etc.

Reviewer #3 (Remarks to the Author):

REVIEWER RECOMMENDATION:

Recommendation: Reconsidering after major revision (as noted below).

REVIEWER COMMENT:

"The Cu₅₀Co₅₀ nanosheet delivers a lowest overpotential of 290 mV for ammonia production, and a 100 ± 1% Faradaic efficiency at an ampere-level current density of 1035 mA cm⁻² for NH₃ production at -0.2 V vs. RHE. The NH₃ production rate reaches 4.8 mmol cm⁻² h⁻¹ (960 mmol g⁻¹ h⁻¹) that is about 5 times higher than the production rate via the Haber-Bosch route." This manuscript outlines evaluating electrochemical nitrate reduction in various conditions with diverse characterizations including in-situ FTIR, SHINERS and DFT calculation. However, some important discussion and information are missing such as the investigation about the stability. I recommend reconsidering this work, please see the comments below.

COMMENT 1:

The author introduced that the enzyme inspired catalysts using the enzyme-metal core but the advantages of using the enzyme-metal core are not clear. Please provide bare CuCo alloy nanosheets or CuCo plates for nitrate reduction and ammonia production without the enzyme-metal core as a reference.

COMMENT 2:

From nitrate to ammonia or nitrogen, the NO^* is a key intermediate product to determine the final products and the catalytic selectivity but there is no investigation or mention of it.

COMMENT 3:

Please provide sufficient results in terms of the stability data how change or maintain the morphology, chemical compositions, and oxidation states with several repeated cycles.

COMMENT 4:

The author insisted the Ni foam is inactivity toward nitrate reduction by providing LSV curves collected by the current density in terms of ECSA in Fig. S4. Please provide the current density by geometric area and how much the concentrations of nitrate ions could be reduced before and after nitrate reduction tests by using only Ni foam.

COMMENT 5:

How precisely control the atomic percent between Cu and Co in this electrodeposition method? Also, please provide all XRD data for Cu₇₀Co₃₀, Cu₅₀Co₅₀, and Cu₂₀Co₈₀.

COMMENT 6:

There is a typo in abstract "The Cu₅₀Co₅₀ nanosheet delivers a lowset overpotential~", revise from "lowset" to "lowest".

Responds to Reviewers

We would thank all the reviewers for their comments and constructive advice, which are helpful to improve the quality of our paper. We have revised our manuscript by carefully considering all comments and constructive advice, and now provide a point-by-point response to the reviewers' comments below. We also have conducted additional analysis to offer more evidence to support our conclusions. To easily see the changes, we highlighted all revisions using red color in the revised manuscript and the Supporting Information.

Reviewer #1:

Comment 1:

In "Abstract", the authors stated that "a $100 \pm 1\%$ Faradaic efficiency at an ampere-level current density of 1035 mA cm^{-2} for NH_3 production at -0.2 V vs. RHE ." However, as shown in Figure 2a, the current density of $\text{Cu}_{50}\text{Co}_{50}$ nanosheet at -0.2 V vs. RHE is $\sim 300 \text{ mA cm}^{-2}$. Please explain why?

Answer 1:

Thank you for your careful review. The current density of 1035 mA cm^{-2} for NH_3 production at -0.2 V vs. RHE was obtained in an H-type cell under magnetic stirring of 1000 rpm. The current density of $\sim 300 \text{ mA cm}^{-2}$ shown in Figure 2a was obtained in a single cell without magnetic stirring. Indeed, the magnetic stirring enhances the NO_3^- mass transfer, which significantly impacts the current density. As shown in Fig. R1a below, the peak current density in LSV curves increased with increasing potential scan rate, indicating that the NO_3^- RR was diffusion-controlled. We also compared the current density of NO_3^- reduction at scan rate 5 mV s^{-1} with and without magnetic stirring in the single cell, and found that the former current density was much larger than the latter. Besides, in the Fig. R1b beneath, the time-dependent current density curves on $\text{Cu}_{50}\text{Co}_{50}$ at different electrode potentials were recorded at a steady state with magnetic stirring speed of 1000 rpm in order to avoid the mass transfer limitation and to maintain a constant NO_3^- ion concentration on electrode, we can see clearly that the current densities obtained on $\text{Cu}_{50}\text{Co}_{50}$ catalyst reach Ampere level at -0.2 V vs. RHE or more negative.

In order to clarify this point, the Fig. R1a was added to the revised manuscript as Fig.2e, and a comparison of time-dependent current density curves on $\text{Cu}_{50}\text{Co}_{50}$, Co and Cu modified Ni foam at -0.2 V vs. RHE with magnetic stirring speed of 1000 rpm was added to the revised manuscript as Fig.2f.

Fig. R1: **a** j - E curves on Cu₅₀Co₅₀ modified Ni foam in 1 M KOH solution containing 100 mM KNO₃ at different scan rate without agitation (solid line) and at scan rate of 5 mV s⁻¹ with agitation. **b** the time-dependent current density curves on Cu₅₀Co₅₀ modified Ni foam at different electrode potential with magnetic stirring speed of 1000 rpm.

Comment 2.

In this work, the authors emphasized the strategy of enzyme-inspired design to boost the activity of nitrate reduction. However, in “Abstract”, the authors did not mention how to achieve enzyme-inspired design.

Answer 2:

We have to sincerely apologize for our previous ambiguous presentation of the enzyme-inspired design strategy. Enzymes are wonders of nature, with specific and efficient catalytic activities for their substrates. It is difficult and not cost-effective to artificially synthesize catalysts mimicking enzymes which are complex systems. Copper-type nitrite reductases (Cu-NIRs) widely found in Rhizobium for nitrogen fixation and are trimeric proteins, which are composed of 3 identical subunits. Each monomer has two types of copper atomic active centers; one behaves as a catalytic center (T2Cu) facilitating *NO₂⁻ adsorption/association and the other acts as an electron/proton donating center (T1Cu) promoting the breaking of the N-O bond. According to the mechanism of nitrite reduction on T2Cu and T1Cu, one can rationally speculate that moderate affinity for NO₃⁻, protons availability and electrons provision are the key factors for effectively reducing NO₃⁻ (NO₃⁻ RR) to NH₃. Based on the functioning mechanism of Cu-NIRs, we have designed and prepared a Cu-based bimetallic catalyst that is able to mimic the behavior of the two catalytic centers of the Cu-NIRs: herein Cu would act as T2Cu and a second metal would act as T1Cu providing the protons and electrons. Co was selected as the second metal due to its activity toward hydrogen evolution reaction with a high hydrogen adsorption energy (*H). Electrochemical *in-situ* FTIR experiments have demonstrated, that protons adsorbed preferentially on Co in Cu₅₀Co₅₀ catalysts (Fig. 4e-g in the manuscript), indicating that Co provides the protons for the NO₃⁻ reduction. Besides, XPS and Extended X-ray absorption fine structure (XANES) spectra, highlighted electrons transfer from Co species to Cu species. The “enzyme-like” of our catalyst is based on its behavior *i.e.*

cooperation between the two active centers in the same way as the ones in the Cu-NIRs.

We have added some descriptions in the “Abstract” to clarify how to achieve the enzyme-inspired design. In addition, the introduction section was well rewritten to clearly explain how we designed the catalysts to obtain a high catalytic ability for NO_3^- RR.

Comment 3.

More gaseous products (i.e., N_2O , NO_2 , NH_3) should be quantified. Especially, part of the generated NH_3 can probably be stripped out from electrolyte in the alkaline solution (1 M KOH).

Answer 3:

We agree that the identification of the potential gaseous intermediates is essential to the mechanistic study. Here we detected the potential gaseous products by online electrochemical Mass Spectrometry (OEMS). The carrier gas is Ar which flows over the electrolyte surface in the course of electrolysis. The curves of the intensity of N_2O , NO_2 , NO and NH_3 vs. time in Fig. R2 illustrated no detection of N_2O , NO_2 and NO . Occasional appearances of NH_3 were observed, ascribing to the slight disturbance by releasing H_2 bubbles. We tried to collect the NH_3 in the tail gas by 1 mol L^{-1} sulfuric acid solution, but no NH_3 was detected. The detection limit of NH_3 measured by the spectrophotometric analysis based on Nessler's reagent is about 5.4×10^{-3} mmol L^{-1} . Therefore, although ammonia in alkaline solutions is easily stripped out, the amount of the stripped NH_3 in this work can be neglected.

Fig. R2: The online electrochemical mass spectrometry result for the possible gaseous products (H_2 , N_2 , NO , NO_2 , N_2O and NH_3) of NO_3^- RR.

Comment 4:

The quantification methods should be at least two types. The authors should refer to the recent advances in the electrochemical reduction of N_2 that introduce how to quantify ammonia. Also, the ammonia yield in electrolyte free of nitrate should also be tested.

Answer 4:

We agree that the precise quantification analysis for NH_3 is important to obtain reliable results, *e.g.*, current density and Faradaic efficiency for NH_3 formation. Spectrophotometric analysis was previously taken to determine the NH_3 concentration in this work. Here, for comparison, we also analyzed the content of $^{14}NH_3$ by 1H NMR test ($4.62 \text{ mmol cm}^{-2} \text{ h}^{-1}$) (Fig. R3d) and obtained a similar conclusion as the spectrophotometric analysis ($4.75 \text{ mmol cm}^{-2} \text{ h}^{-1}$), which confirmed that the NH_3 concentrations we reported in the text were reliable.

Fig. R3: 1H NMR spectra of different concentrations of $^{14}NH_4Cl$ (a) and $^{15}NH_4Cl$ (b) standard solution and the calibration curves of normalized integral area between NH_4^+ and $C_4H_4O_4$ vs. the concentration of NH_4^+ (c). **d** The $^{14}NH_3$ yield rate and Faradaic efficiency detected by 1H NMR spectroscopy and the Nessler reagent method at -0.2 V vs. RHE. **e** 1H NMR spectrum of the electrolyte after the electrolysis of $^{14}NO_3^-$ and $^{15}NO_3^-$ at -0.2 V vs. RHE. **f** The $^{15}NH_3$ yield rate and Faradaic efficiency detected by 1H NMR spectroscopy and the Nessler reagent method at -0.2 V vs. RHE.

As shown in Fig. R4, the current density for NH_3 formation in the solution with the presence of NO_3^- ions is over 150-fold higher than in its absence, where the trace amount of NH_3 might come from the N_2 reduction. Hence, the N_2 reduction reaction has little effect on the quantification of NH_3 produced by NO_3^- RR. In addition, the typical two peaks of $^{15}NH_4^+$ after the electrolysis of $^{15}NO_3^-$ also suggested that the NH_3 product indeed came from the electrocatalytic reduction of NO_3^- (Fig.

R3e).

Fig. R4: Current density and yield rate for NH_3 for $\text{Cu}_{50}\text{Co}_{50}/\text{Ni}$ foam at -0.2 V vs. RHE in 1 M KOH electrolyte with or without 100 mM KNO_3 .

Comment 5:

More experimental data are expected to support the durability of the catalysts. How about the variation in the performance of ammonia yield and Faradaic efficiency during the consecutive recycling test? Were the catalysts stable during the nitrate reduction? More characterizations and comparisons are needed for the catalysts before and after the measurements.

Answer 5:

Indeed, the durability of the catalysts is another important indicator of catalytic performance. Ten cycle stability tests were performed on $\text{Cu}_{50}\text{Co}_{50}$ in an H-type cell (Fig. R5). The Faraday efficiency remained stable and were more than 90%. Although the $\text{Yield}_{\text{NH}_3}$ decreased gradually to half, which was also two times larger than the Haber–Bosch process (the red dash line in Fig. R5). According to SEM analysis, the decay of the $\text{Yield}_{\text{NH}_3}$ might be due to the nanosheet agglomeration after the consecutive recycling tests (the red cycle in Fig. R6b). XRD (Fig. R6c) and XPS analysis (Fig. R6d and e) were also done on the samples after the consecutive recycling tests. The results demonstrated negligible changes of the chemical compositions and oxidation states, which further confirmed the good stability of the catalyst.

Fig. R5: FE and Yield rate of NH_3 on $\text{Cu}_{50}\text{Co}_{50}/\text{Ni}$ foam under the applied potential of -0.2 V vs. RHE during 10 periods of 1 h electrocatalytic NO_3^- RR. (The red dash line was the $\text{Yield}_{\text{NH}_3}$ of Haber-Bosch process.)

Fig. R6: SEM images of the $\text{Cu}_{50}\text{Ni}_{50}$ catalyst before (a) and after 10 periods of 1 h (b) electrocatalytic NO_3^- RR. XRD spectra (c) and XPS peaks spectra of Cu 2p (d) and Co 2p (e) of $\text{Cu}_{50}\text{Co}_{50}$ catalyst before and after 1 h, 10 periods of 1 h NO_3^- RR operation at -0.2 V vs. RHE .

Comment 6:

Electrochemical nitrate reduction in an alkaline electrolyte involves eight electrons transfer coupled with the generation of nine OH^- , so what was the H^+ source of the final NH_3 product? The pH change during the nitrate reduction in alkaline media should be monitored.

Answer 6:

Due to the lack of experimental equipment, we couldn't achieve real-time pH monitoring. However, we tested the pH of the electrolyte before and after 1 hour of electrolysis at -0.2 V vs. RHE. It was found that the pH value increased from 13.61 to 13.68, indicating that the consumption of H came from H_2O dissociation in the process of NO_3^- reduction, leading to the formation and accumulation of OH^- . We speculated that the hydrogen proton of the final NH_3 product originated from the H_2O dissociation. As there was the ion exchange of Nafion117 membrane, which might affected the pH detecting. In order to get direct experimental evidence, we also performed the NO_3^- reduction in D_2O solvent at -0.2 V vs. RHE for 1 h, and analyzed the m/z of formed ammonium by electrochemical mass spectrometry. The carrier gas was Ar that flowed through the bottom of the electrolyte to the surface. The electrolyte was kept at 60 °C to lower the solubility of ammonia in the solution. We found that ND_3 was the only product (Fig. R7), which confirmed that dissociation water provided the H^+ for NO_3^- and its derivatives hydrogenation.

Fig. R7: The electrochemical mass spectrometry result for ND_3 , NHD_2 , NH_2D and NH_3 in electrolyte at -0.2 V vs. RHE after 1 hour NO_3^- RR.

Comment 7:

Considering that the pH of waste water is mostly ~ 7 , the authors should test the performance of different electrocatalysts in neutral media.

Answer 7:

The NO_3^- RR over $\text{Cu}_{50}\text{Co}_{50}$, Cu and Co catalysts in neutral conditions was tested at -0.2 V vs. RHE in the electrolyte of 0.1 M KNO_3 + 0.5 M K_2SO_4 (pH around 6.82). As shown in Fig. R8, the highest current density of NO_3^- RR in a neutral condition was obtained over $\text{Cu}_{50}\text{Co}_{50}$. FE_{NH_3} over $\text{Cu}_{50}\text{Co}_{50}$ and Co were much higher than Cu, which is the same in an alkaline electrolyte. Therefore, it can be confirmed that incorporating the Co element is beneficial to reduce the intermediate

product, e.g., nitrite, independent of solution pH. The pH of the neutral electrolyte increased from 6.82 to 11.41 after NO_3^- RR for 1 hour. On the other hand, we also conducted the removal of $200 \text{ mg L}^{-1} \text{NO}_3^-$ in neutral conditions that is close to the concentration of NO_3^- pollutant in wastewater¹, and found that the highest NO_3^- removal rate of 98% was also obtained on $\text{Cu}_{50}\text{Co}_{50}$ catalyst, and the residual NO_3^- concentration was less than 5 mg L^{-1} .

Fig. R8: Current density and FE_{NH_3} and $FE_{\text{NO}_2^-}$ of NO_3^- RR in $100 \text{ mM KNO}_3 + 0.5 \text{ M K}_2\text{SO}_4$ neutral electrolyte (a) and NO_3^- removal rate after 2 hour reduction in $0.1 \text{ M K}_2\text{SO}_4$ electrolyte with 200 mg NO_3^- (b) on Cu, $\text{Cu}_{50}\text{Co}_{50}$ and Co catalysts at -0.2 V vs. RHE .

Comment 8:

After alloying with Co, the lattice spacing of Cu (111) plane is contracted. In general, the lattice constrain of metal sites tunes the adsorption strength towards reaction intermediates and thus the reaction activity. Did the lattice constrain of Cu also impact the activity?

Answer 8:

We agree that the lattice constrains of Cu would impact the activity of NO_3^- RR. Meanwhile, we discovered that incorporating Co with a smaller atomic radius into the Cu(111) lattice would lead to the contraction of Cu lattice spacing, which was confirmed by the XRD and SEM analysis (Figure 1b, c and Supplementary Fig. S1a, b). DFT calculation (Figure 5a and Supplementary Table S1) also proved that CuCo(111) has greater adsorption energy for NO_3^- than Cu(111), which corresponds to a higher activity of NO_3^- RR on the former plane. Besides, as can be seen from the DFT calculation of NH_3 desorption steps, the Gibbs free energy of $^*\text{NH}_3$ desorption on CuCo(111) is higher than that on Cu(111), which may be caused by the enhancement of the adsorption energy of intermediates by the shrinkage of lattice spacing inside of CuCo(111). Fortunately, the $^*\text{NH}_3$ desorption is not the rate-determining step, which less impacts the overall reaction. Based on the analysis of the existing experimental results, the shrinkage of the Cu lattice spacing will strengthen the adsorption of reaction intermediates. Appropriate enhancement of the adsorption to intermediate species enhances the catalytic activity of NO_3^- reduction to NH_3 production.

Reviewer #2 (Remarks to the Author).

Comment 1:

I don't particularly understand what is bio-inspired or enzyme-like about these catalysts. A hallmark of enzyme catalysis is the specificity; this would not seem to be the case here. I felt a rather weak case was made when trying to bridge Cu-containing reductases with metallic catalysts. The paper may be improved by eliminating this biological comparison, if no stronger connections to enzyme form/function can be made. Relatedly, the nature of this catalyst is still unclear to me. The authors describe these as CuCo sheets, though the images in Figure 1 and Figure S2 seem to suggest otherwise. If they are sheets, how does that relate to the enzyme-like behavior? A more clear and consistent description of the catalyst would help.

Answer 1:

We have to sincerely apologize for our previous ambiguous presentation of the enzyme-inspired design strategy. Enzymes are wonders of nature, with specific and efficient catalytic activities for their substrates. It is difficult and not cost-effective to artificially synthesize catalysts mimicking enzymes which are complex system. Copper-type nitrite reductases (Cu-NIRs) widely found in *Rhizobium* for nitrogen fixation and are trimeric proteins, which are composed of 3 identical subunits. Each monomer has two types of copper atomic active centers; one behaves as a catalytic center (T2Cu) facilitating $^*NO_2^-$ adsorption/association and the other acts as an electron/proton donating center (T1Cu) promoting the breaking of the N-O bond. According to the mechanism of nitrite reduction on T2Cu and T1Cu, one can rationally speculate that moderate affinity for NO_3^- , protons availability and electrons provision are the key factors for effectively reducing NO_3^- ($NO_3^- \rightarrow RR$) to NH_3 . Based on the functioning mechanism of Cu-NIRs, we have designed and *vs.* RHE prepared a Cu-based bimetallic catalyst that is able to mimic the behavior of the two catalytic centers of the Cu-NIRs: herein Cu would act as T2Cu and a second metal would act as T1Cu providing the protons and electrons. Co was selected as the second metal due to his activity toward hydrogen evolution reaction with a high hydrogen adsorption energy (*H). Electrochemical *in-situ* FTIR experiments have demonstrated, that protons adsorbed preferentially on Co in $Cu_{50}Co_{50}$ catalysts (Fig. 4e-g in the manuscript), indicating that Co provides the protons for the NO_3^- reduction. Besides, XPS and Extended X-ray absorption fine structure (XANES) spectra, highlighted electrons transfer from Co species to Cu species. The “enzyme-like” of our catalyst is based on its behavior *i.e.* cooperation between the two active centers in the same way as the ones in the Cu-NIRs.

We have added some descriptions in the “Abstract” to clarify how to achieve the enzyme-inspired design. In addition, the introduction section was well rewritten to clearly explain how we designed the catalysts to obtain a high catalytic ability for NO_3^-RR .

Comment 2:

The authors compare NH_3 production rates on a mass basis with the Haber-Bosch catalyst. I'm not sure this is an appropriate comparison, given the highly disparate costs of Cu and Fe catalysts, as

well as the major differences between a lab-scale electrochemical setup and the industrial scale Haber-Bosch process.

Answer 2:

We agree that it is not an appropriate comparison between the electrosynthesis of NH_3 with the Haber-Bosch method at now, considering the huge differences between a lab-scale and the industrial scale. Therefore, the previous title “Enzyme-Inspired CuCo nanosheet for electrochemical synthesis of Ammonia as a potential substitute of Haber-Bosch method” has been changed to “Enzyme-Inspired CuCo Nanosheet for Electrochemical Synthesis of Ammonia at an Ampere-Level Current Density”. Currently, the Haber-Bosch process is a well-developed method of synthetic ammonia on an industrial scale that is responsible for the main ammonia supply world widely. The $\text{Yield}_{\text{NH}_3}$ of the electrocatalytic NO_3^- RR at unit mass catalyst in this work achieves to $960 \text{ mmol g}_{\text{cat}}^{-1} \text{ h}^{-1}$ at -0.2 V vs. RHE , which is indeed superior to the $\text{Yield}_{\text{NH}_3}$ in the Haber-Bosch method ($200 \text{ mmol g}_{\text{cat}}^{-1} \text{ h}^{-1}$).² We agree that the production of NH_3 from electrocatalytic NO_3^- RR reported in this paper is only on a laboratory scale and there are still many follow-up pilot tests and scale-up work to meet the industrial demands, and it provides nevertheless a new expectation and shows a great prospect.

Comment 3:

What is the role of the Ni foam? Would this process work if simply depositing Co on a Cu foam?

Answer 3:

Ni foam is widely used as supporting substrate for nanostructured electrocatalysts in terms of the smooth surface that is suitable for electrodeposition³⁻⁵. Meanwhile, thanks to its good conductivity, which benefits the efficient electron transfer. Ni is proved as a relatively inert material for NO_3^- RR^{6,7}, which is beneficial for us to focus on the investigation of the performance of the CuCo catalysts, which mimic the active sites of enzyme.

After replacing Ni foam with Cu foam as substrate, we found that the FE_{NH_3} close to 100% could be maintained (Fig. R9). The corresponding NH_3 yield rate over $\text{Cu}_{50}\text{Co}_{50}/\text{Cu}$ foam was *ca.* $5.42 \text{ mmol h}^{-1} \text{ cm}^{-2}$, which is even higher than that on $\text{Cu}_{50}\text{Co}_{50}/\text{Ni}$ foam ($4.80 \text{ mmol h}^{-1} \text{ cm}^{-2}$) since the Cu substrate has a good catalytic activity for NO_3^- RR. In the case of carbon paper used as the substrate, which also achieves a 100 % FE_{NH_3} . But the NH_3 yield rate over $\text{Cu}_{50}\text{Co}_{50}/\text{CP}$ ($4.15 \text{ mmol h}^{-1} \text{ cm}^{-2}$) was slightly lower than the results on metallic substrates, which was probably due to the relatively poor electrical conductivity. The NO_3^- RR on Co/Cu foam was also conducted. We found a significant lower productivity of NH_3 ($1.85 \text{ mmol h}^{-1} \text{ cm}^{-2}$) than that on the $\text{Cu}_{50}\text{Co}_{50}/\text{Cu}$ foam, though the FE_{NH_3} was close to 100%.

Fig. R9: FE_{NH_3} of NO_3^- RR and product yield rate for NH_3 for $\text{Cu}_{50}\text{Co}_{50}/\text{Ni foam}$, $\text{Cu}_{50}\text{Co}_{50}/\text{Cu foam}$, $\text{Cu}_{50}\text{Co}_{50}/\text{CP}$ and $\text{Co}/\text{Cu foam}$ at -0.2 V vs. RHE in $100 \text{ mM KNO}_3 + 1 \text{ M KOH}$ electrolyte.

Comment 4:

The lattice constants used in calculations should be provided. What is the unit cell size? (This should not need to be inferred from SI figures.) Which atoms were relaxed or fixed in the slabs? What k-point sampling was used? Were calculations performed with spin polarization? Was a dipole correction included for the slab calculations?

Answer 4:

Thank you for the careful review. We apologize for the confusion caused by the lack of detailed instructions. We have added detailed descriptions related to this question in the supporting information and marked them in red and quoted below:

“The converged unit cell models of Cu ($3.64 \text{ \AA} \times 3.64 \text{ \AA} \times 3.64 \text{ \AA}$), Co ($3.52 \text{ \AA} \times 3.52 \text{ \AA} \times 3.52 \text{ \AA}$) and CuCo ($3.78 \text{ \AA} \times 3.49 \text{ \AA} \times 3.49 \text{ \AA}$) were used in DFT calculations, respectively. The dimension of supercell of Cu (111) ($8.91 \text{ \AA} \times 10.28 \text{ \AA} \times 24.20 \text{ \AA}$), Co (111) ($8.62 \text{ \AA} \times 9.95 \text{ \AA} \times 19.06 \text{ \AA}$) and CuCo (111) ($9.03 \text{ \AA} \times 9.87 \text{ \AA} \times 19.13 \text{ \AA}$) were used, respectively. These supercells were constructed and contained three layers and a sufficient vacuum layer of 15 \AA thicknesses. The two layers on the bottom were fixed, and the top layer was fully relaxed⁸. For unit cell geometry optimization, an $8 \times 8 \times 8$ k-point analysis was used. For these supercell calculations, a grid of $3 \times 3 \times 1$ k-point mesh was used³. Dipole corrections in the z direction were included in all computations to minimize inaccuracies in the total energy because of simulated slab interactions. The calculations of all molecules and intermediate species on Co (111) and CuCo (111) were performed with spin polarization.⁹ We found that spin polarization does not affect the energy of intermediate species on Cu (111) (Table R1). Therefore, considering the limited computational resources, the calculations of intermediate species on Cu (111) do not consider the spin polarization.”

Table R1: the total energy of $^*\text{NO}_3$ and $^*\text{NO}_2$ on Cu(111) performed with or without spin

polarization

Cu(111)	with spin polarization (eV)	without spin polarization (eV)
*NO ₃	-188.29	-188.29
*NO ₂	-182.66	-182.66

Comment 5:

How are free energies calculated, and what are the reference states? Relatedly, how was the adsorption energy of the negatively charged species modeled?

Answer 5:

The NO₃⁻RR were simulated according to the following reactions, Eqs. (1-9):

where the * represent the active sites. The adsorption energy of intermediates on different catalyst's surfaces was calculated by Eq. (10):

$$\Delta E_{\text{ads}} = E_{*A} - E_* - E_A \quad (10)$$

where E_{*A} , E_* and E_A denote the total energy of the adsorbed system, the clear substrate and adsorbate, respectively.

In thermodynamic calculation, the total energy of negatively charged species like NO₃⁻ and OH⁻ are difficult to deal with. For this reason, we used the stable molecules HNO₃, H₂O and H₂ instead. The free energies of HNO₃, H₂O and H₂ were used as references when calculating the free energies of reaction intermediates, following Eqs. (11) and (12).

$$E(\text{OH}^-) = E(\text{H}_2\text{O}) - \frac{1}{2}E(\text{H}_2) \quad (11)$$

$$\text{TS}(\text{OH}^-) = \text{TS}(\text{H}_2\text{O}) - \frac{1}{2}\text{TS}(\text{H}_2) \quad (12)$$

For Eq. (1), we applied 0.75 eV correction (containing entropic and enthalpic contributions) to compensate for the DFT calculation^{10,11}. The free energy of nitrate adsorption from the solution phase vs reverse hydrogen electrode is presented by Eq. (13).

$$\Delta G_{\text{NO}_3} = E(*\text{NO}_3) + \frac{1}{2}E(\text{H}_2) - E(*) - E(\text{HNO}_3) + 0.75 \text{ eV} \quad (13)$$

For each subsequent reaction, the free energies were given after gas correction, following Eq. (14):

$$\Delta G = \Delta E + \Delta \text{ZPE} - T\Delta S - eU \quad (14)$$

where ΔE is the energy obtained by the difference between reactant and product, ΔZPE denotes the change of zero-point energy. ΔS is the change in entropy for each reaction. The entropies of adsorbate and adsorption sites are negligible. Here, U is the potential at the electrode and e is the transferred charge. Enthalpic contribution of gas corrections for H_2O , H_2 and NH_3 used in the Gibbs free energy calculations was listed in Table R2.

By this approach, taking Eq. (3) as an example, the Gibbs free energy is calculated by

$$\begin{aligned} \Delta G_3 &= \Delta E - T\Delta S \\ &= E(*\text{NO}) + 2E(\text{OH}^-) - E(*\text{NO}_2) - E(\text{H}_2\text{O}) - [2\text{TS}(\text{OH}^-) - \text{TS}(\text{H}_2\text{O})] \\ &= E(*\text{NO}) + 2[E(\text{H}_2\text{O}) - \frac{1}{2}E(\text{H}_2)] - E(*\text{NO}_2) - E(\text{H}_2\text{O}) - [2(\text{TS}(\text{H}_2\text{O}) - \frac{1}{2}\text{TS}(\text{H}_2)) \\ &\quad - \text{TS}(\text{H}_2\text{O})] \\ &= E(*\text{NO}) + E(\text{H}_2\text{O}) - E(\text{H}_2) - E(*\text{NO}_2) - [\text{TS}(\text{H}_2\text{O}) - \text{TS}(\text{H}_2)] \end{aligned}$$

Table R2: Enthalpic contribution of gas corrections for H_2O , H_2 and NH_3 used in the Gibbs free energy calculations. The room temperature $T = 298.15 \text{ K}$.

Molecule	$T\Delta S$	Ref.
H_2O	0.67 eV	12
H_2	0.41 eV	12
NH_3	0.60 eV	13

Comment 6:

All calculated energies for intermediates should be provided in SI tables.

Answer 6:

This is a very good suggestion for reader-friendly. We added the Table of calculated energies for intermediates shown below to SI and marked it in red.

Table R3: The correction of zero-point energy of adsorption species and molecules involved in reaction

	Cu(111)	Co(111)	CuCo(111)
*NO ₃	0.79	0.92	0.89
*NO ₂	0.66	0.81	0.75
*NO	0.56	0.68	0.68
*NOH	0.86	0.96	0.97
*NHOH	1.18	1.29	1.27
*NH ₂ OH	1.49	1.63	1.59
*NH ₂	1.07	1.20	1.17
*NH ₃	1.38	1.53	1.50
*	0.39	0.52	0.49
*H	0.56	0.69	0.67
H ₂ (g)	0.27	0.27	0.27
H ₂ O (g)	0.57	0.57	0.57
NH ₃ (g)	0.91	0.91	0.91
HNO ₃ (l)	0.69	0.69	0.69

Table R4: Calculated total energies (in eV) on different catalysts surfaces

	Cu(111)	Co(111)	CuCo(111)
*	-162.86	-317.95	-233.09
*NO ₃	-188.29	-343.62	-258.94
*NO ₂	-182.66	-338.01	-252.94
*NO	-176.57	-332.66	-247.60
*NOH	-179.97	-335.96	-251.05
*NHOH	-183.84	-339.45	-254.66
*NH ₂ OH	-187.60	-342.90	-258.09
*NH ₂	-178.82	-334.23	-249.18
*NH ₃	-182.92	-338.19	-253.36

*H	-166.57	-321.82	-236.97
----	---------	---------	---------

Table R5: Calculated Gibbs free energies (in eV) on different catalysts surfaces

	Cu(111)		Co(111)		CuCo(111)	
	U = 0 V	U = -0.2 V	U = 0 V	U = -0.2 V	U = 0 V	U = -0.2 V
ΔG_1	0.38	0.58	0.14	0.34	-0.03	0.17
ΔG_2	-1.90	-2.30	-1.91	-2.31	-1.54	-1.94
ΔG_3	-1.41	-1.81	-2.17	-2.57	-2.14	-2.54
ΔG_4	0.36	0.16	0.43	0.23	0.30	0.10
ΔG_5	-0.09	-0.29	0.29	0.09	0.14	-0.06
ΔG_6	0.00	-0.20	0.35	0.15	0.36	0.16
ΔG_7	-2.49	-2.69	-2.62	-2.82	-2.38	-2.58
ΔG_8	-0.33	-0.53	-0.16	-0.36	-0.39	-0.59
ΔG_9	-0.16	-0.16	0.00	0.00	0.04	0.04

Comment 7:

How does NO_3 dissociation occur? If it is assumed to occur by direct N-O bond scission, there is an energy barrier associated with this that may be larger than other calculated free energy barriers. The implications of activation energy barriers should be discussed, if not explicitly calculated.

Answer 7:

According to the work of Guo and his colleagues¹⁰, there are two kinds of NO_3^- RR pathways on Cu reported in the literature^{10,14}, including cleavage of the N-O bond after hydrogenation and N-O bond breaking directly (Fig. R10). On Cu (111), the activation energy barrier for the cleavage of the N-O bond after hydrogenation ($^*\text{NO}_3 \rightarrow ^*\text{NO}_3\text{H} \rightarrow ^*\text{NO}_2$, 1.71 eV) is greater than that of the N-O bond breaking directly (0.68 eV).¹⁰ The dissociation reaction of NO_3^- needs to be explained in terms of kinetics.

Fig. R10. (a)The activation energy data of the reaction kinetics on Cu(111) for NO_3^- RR. (given in eV) (b) Potential energy diagrams, and transition states for the conversion of NO_3 to NH_3 on Cu (111) surface. Cited from reference¹⁰ (Guo et al, ACS Catal. 2021).

The reaction pathway consists of a series of reaction intermediates with specific sequences under thermodynamics. The identification of the reaction pathway is vital to understand the NO_3^- RR. When discussing the most favorable reaction pathways, it is necessary to identify the activation energy barrier between each sequence step. In this work, we experimentally identified the intermediate species by in-situ FTIR (Fig. 4 in the manuscript) and SHINERS (Fig. 5 in the manuscript). That is a series of deoxygenation, $\text{NO}_3^- \rightarrow * \text{NO}_3 \rightarrow * \text{NO}_2 \rightarrow * \text{NO}$, and accompanied by a series of hydrogenation: $* \text{NO} \rightarrow * \text{NOH} \rightarrow * \text{NH}_2\text{OH} \rightarrow * \text{NH}_3 \rightarrow \text{NH}_3$.

Besides, we focused on the difference in catalytic activities of NO_3^- RR between these different catalysts. Therefore, we have only analyzed the NO_3^- RR pathways in terms of thermodynamics. The sentence “The rate-determining step (RDS) on” has now been rewritten as " In terms of thermodynamics, the rate-determining step (RDS) on indicating a stronger adsorption of NO_3^-

species on CuCo(111)." for a clearer description and the change was marked in red in second paragraph of the section of **DFT**, page 18.

Comment 8:

Why was an fcc model used for Co, which has an hcp structure? I was expecting to see Co (0001) used.

Answer 8:

We apologize for the missing explanation of the choice of Cu, Co and CuCo models in previous manuscript. We used the fcc model for Co based on the XRD results, which showed that Cu, Co and CuCo catalysts were fcc crystal structures (Figure 1b in the text). Limited by computational resources, NO₃RR on Co (0001) has not yet been calculated.

Comment 9:

How was the bimetallic constructed, including determination of an appropriate lattice constant? Is there experimental evidence for alternating rows of Co and Cu? The authors may be missing important features about potential active sites if restricting things to these highly ordered sites. Limitations should be briefly discussed.

Answer 9:

We appreciate your helpful questions and suggestion again.

The XRD pattern showed that the peak of CuCo(111) was between Cu and Co, which indicated the lattice shrinkage of Cu caused by the introduction of Co. The lattice constant of the crystal plane of Cu(111), Co(111) and CuCo(111) was listed in Table R6. The theoretically calculated values were close to the lattice constant of metal in standard Powder Diffraction File and the trend of calculated lattice change was consistent with the XRD results. Therefore, The CuCo model we built matched the experimental analysis.

Table R6: The lattice constant of the crystal plane of Cu, Co and CuCo alloys (111)

	XRD results	Theoretical calculation
Cu(111)	2.09 Å (PDF#04-0836)	2.10 Å
CuCo(111)	/	2.07 Å
Co(111)	2.05 Å (PDF#15-0806)	2.03 Å

We heartily agree with the concern about the consistency of the bimetallic model with the experiment. At present, it is difficult to visually "see" the realistic arrangement of atoms in CuCo alloy experimentally. We wish to demonstrate the rationality and limitations of the bimetallic model used in this work from the supplementary computational results. We also agree that the adsorption mode between intermediate species and surface sites is critical to catalytic activity. Therefore, 5 new CuCo(111) modes with different surface atomic arrangements of Cu and Co were designed complementally.

(1) Construction of bimetallic model:

Firstly, we replace half of the Cu atoms with Co atoms in the converged Cu unit cell ($3.64 \text{ \AA} \times 3.64 \text{ \AA} \times 3.64 \text{ \AA}$). The converged CuCo unit cell ($3.78 \text{ \AA} \times 3.49 \text{ \AA} \times 3.49 \text{ \AA}$) was obtained after geometry optimization. Based on this CuCo unit cell, we constructed the three layers of a supercell with alternating rows of Co and Cu atoms. Then the two layers on the bottom were fixed while the top layer was fully relaxed. Secondly, for supercell geometry optimization, all atoms are relaxed. All atoms are fixed in the intermediate species calculations on the model's surface. In the calculations of intermediate species, the two layers on the bottom of the supercell were fixed while the top layer was fully relaxed.

(2) To study the adsorption mode of intermediate species and active sites, we studied the process of $* + \text{NO}_3^- \rightarrow *\text{NO}_3 \rightarrow *\text{NO}_2$ on the 5 new added CuCo modes, since $* + \text{NO}_3^- \rightarrow *\text{NO}_3$ is the rate-determining step. For the reason of computing resources limitation, the following processes were omitted. The models were shown below (Table R7):

Table R7: Top view and side view of Cu(111), Co(111) and CuCo(111) with different surface exposure structures and the corresponding adsorption configuration of $*\text{NO}_3$ and $*\text{NO}_2$

		Cu	CuCo	CuCo-1	CuCo-2	CuCo-3	CuCo-4	CuCo-5	Co
slab	top view									side view								$*\text{NO}_3$									$*\text{NO}_2$									
Table R8: Calculated Gibbs free energies (in eV) of $*\text{NO}_3$ and $*\text{NO}_2$ on Cu(111), Co(111) and CuCo(111) with different surface exposure structures at 0.0 V vs. RHE.

	Cu	CuCo	CuCo-1	CuCo-2	CuCo-3	CuCo-4	CuCo-5	Co
$\Delta G1$	0.38	-0.03	0.35	0.32	0.31	0.10	0.18	0.14
$\Delta G2$	-1.90	-1.54	-1.79	-1.90	-1.93	-1.91	-1.94	-1.91

Table R9: Calculated Gibbs free energies (in eV) of *NO₃ and *NO₂ on Cu(111), Co(111) and CuCo(111) with different surface exposure structures at -0.2 V vs. RHE.

	Cu	CuCo	CuCo-1	CuCo-2	CuCo-3	CuCo-4	CuCo-5	Co
ΔG1	0.58	0.17	0.55	0.52	0.51	0.30	0.38	0.34
ΔG2	-2.30	-1.94	-2.19	-2.30	-2.33	-2.31	-2.34	-2.31

The Gibbs free energies of *NO₃ and *NO₂ on Cu(111), Co(111) and CuCo(111) with a different arrangement of surface atoms, denoted as CuCo and CuCo1 to 5, were calculated and presented in Table R8 and R9.

The atom's arrangement on the first layer of the alloy directly affects the adsorption of intermediates. We first compared the ΔG1 of Cu and CuCo-1, which had the same surface composition and active sites for adsorption. The ΔG1 on CuCo-1 was slightly lower than that on Cu, indicating that the alloying between Cu and Co benefits the NO₃⁻RR. As the appearance and gradually increase of Co atoms ratio in the first layer (CuCo-2, 3, 4 and CuCo), the ΔG1 required for NO₃⁻ adsorption gradually decreased from 0.55 eV to 0.17 eV in the case of -0.2 V and from 0.35 eV to -0.03 eV. It demonstrated that the surface layer element composition significantly affected the adsorption mode. However, it should be noted that when the surface layer composes only Co atoms, the adsorption energy of NO₃⁻ will increase. The trend was consistent with the experimental result that $\eta_{10 \text{ mA cm}^{-2}}$ of NO₃⁻RR on Co catalyst was 690 mV that is more negative than on Cu₅₀Co₅₀ catalyst ($\eta_{10 \text{ mA cm}^{-2}}$ of 498 mV) (Fig. 2a in the manuscript). It should be mentioned that a dramatically higher j_{NH_3} was obtained on Cu₅₀Co₅₀ catalyst at 0.0 V (347 mA cm⁻²) compared to Cu catalyst (34 mA cm⁻²) and Co catalyst (21 mA cm⁻²) alone (Supplementary Fig. S24a in the SI). The CuCo model provided more consistent results of DFT calculation with the above experimental results. Therefore, we selected the CuCo model for subsequent analysis to explain the difference between the Cu₅₀Co₅₀ alloy and the pure Cu and Co metal catalysts.

Comment 10:

The authors do not specify an electrochemical model, which is typically required to get accurate insights into reaction energetics and identify quantities such as the rate determining step in electrochemical studies. If the authors intended to provide energetics at 0.0 V vs. RHE and have used H₂ gas as a reference, they may have serendipitously ended up with energetics and conclusions that would be obtained with, e.g., Norskov's computational hydrogen electrode. Still, such a model should be applied and the potential used to calculate reaction energetics should be specified with all corresponding figures. (I might suggest presenting results at -0.2 V vs. RHE, if that is indeed the most interesting potential experimentally.)

Answer 10:

Thanks for your kind guide. We have supplemented the Calculation of Gibbs free energies (in eV) at -0.2 V vs. RHE on different catalysts surfaces. Though we could know from Table R8 and 9 that the ΔG at 0 V vs. RHE and -0.2 V vs. RHE were different, the drift of Gibbs free energy in the NO_3^- RR route was similar (Table R10). The rate-determining step (RDS) on CuCo(111) and Co(111) was the hydrogenation of *NO to *NOH at 0 V vs. RHE while it changed to the adsorption of NO_3^- if calculation at -0.2 V vs. RHE. The reaction free energies for different intermediates on three catalysts were given at -0.2 V vs. RHE in the manuscript and corrected discussions were marked in red on Page 18.

Table R10: Calculated Gibbs free energies (in eV) on different catalysts surfaces at 0 V vs. RHE and -0.2 V vs. RHE

	Cu(111)		Co(111)		CuCo(111)	
	U = 0 V	U = -0.2 V	U = 0 V	U = -0.2 V	U = 0 V	U = -0.2 V
ΔG_1	0.38	0.58	0.14	0.34	-0.03	0.17
ΔG_2	-1.90	-2.30	-1.91	-2.31	-1.54	-1.94
ΔG_3	-1.41	-1.81	-2.17	-2.57	-2.14	-2.54
ΔG_4	0.36	0.16	0.43	0.23	0.30	0.10
ΔG_5	-0.09	-0.29	0.29	0.09	0.14	-0.06
ΔG_6	0.00	-0.20	0.35	0.15	0.36	0.16
ΔG_7	-2.49	-2.69	-2.62	-2.82	-2.38	-2.58
ΔG_8	-0.33	-0.53	-0.16	-0.36	-0.39	-0.59
ΔG_9	-0.16	-0.16	0.00	0.00	0.04	0.04

Comment 11:

On Page 18, the authors discuss a potential redistribution of electrons in $\text{Cu}_{50}\text{Co}_{50}$ and subsequent effects on reaction kinetics. The rationale for this statement is unclear, as I don't see computational evidence for such a redistribution of electrons nor any calculated kinetic quantities.

Answer 11:

We acknowledge your valuable comments. On page 5, according to XPS analysis, we found a notable decrease of the Cu 2p binding energy in CuCo, compared with pure Cu (Supplementary Fig. S4a in the manuscript) and an increase of the Co 2p binding energy, compared with pure Co (Supplementary Fig. S4b in the manuscript). The binding energy shifts revealed a redistribution of the electrons between Cu and Co after alloying¹⁵, leading to the movement of the d band towards the Fermi level¹⁶. Besides, this shift can affect the adsorption energy of *H, *NO₃ and the *NO_x intermediates¹⁷. Here, we also performed the X-ray absorption spectroscopy (XAS) analysis to check the redistribution of electrons between Cu and Co, and the results was shown in Fig. R11. Extended X-ray absorption fine structure (XANES) spectra depicted a negative shift of the absorption edge position was found for the Cu K-edge after interacting with Co elements

compared with that of Cu foil (indicated by the red arrow in the inset), illustrating the electron density transfer from Co to Cu (Fig. R11a). The opposite shift of Co K-edge with introducing Cu elements (Fig. R11b) approves the same conclusion.

In addition, we also calculated the Bader charge of Cu and Co centers in CuCo(111), Cu(111) and Co(111).¹⁸ According to Bader charge analysis (Fig. R12), compared with Cu and Co, there was charge redistribution on CuCo(111), in which Cu center was electrons-rich compared to Co center (Fig. R13).

Fig. R11. Extended X-ray absorption fine structure (XANES) spectra of Cu (a) and Co (b) K-edge of CuCo, compared to the metallic Cu and Co foil used as references. Cu (c) and Co (d) K-edge FT-EXAFS spectra.

Fig. R12: Bader analysis for Cu(111), Co(111) and CuCo(111).

Fig. R13: 2D atomic electron density differences of CuCo(111)

Comment 12:

In Figures 6 and S22, can different colors be used for Cu and Co? It is difficult to distinguish the two.

The writing quality could be improved, particularly in the methods section. There are places where the meaning could be confused: the authors mix the use of “absorption” and “adsorption”, which have different meanings. There is a typo even in the title of the article: “Haber-Bosch”

It is unusual that the methods section was written without any external references; there are typically references provided to previously-used techniques, computational codes, etc.

Answer 12:

The color of Cu has been changed to yellow. We carefully rewrite the methods section to make it clear to read. Sorry for the mistaken use of adsorption and absorption. We have corrected the words in the text and marked them in red. The typo “Harber-Bosch” has been corrected as “Haber-Bosch”. The related references for the previously-used technique, computational codes, etc., have been added to the supporting information.

Reviewer #3 (Remarks to the Author):

Comment 1:

The author introduced that the enzyme inspired catalysts using the enzyme-metal core but the advantages of using the enzyme-metal core are not clear. Please provide bare CuCo alloy nanosheets or CuCo plates for nitrate reduction and ammonia production without the enzyme-metal core as a reference.

Answer 1:

We have to sincerely apologize for our previous ambiguous presentation of the enzyme-inspired design strategy. Enzymes are wonders of nature, with specific and efficient catalytic activities for their substrates. It is difficult and not cost-effective to artificially synthesize catalysts mimicking enzymes which are complex system. Copper-type nitrite reductases (Cu-NIRs) widely found in Rhizobium for nitrogen fixation and are trimeric proteins, which are composed of 3 identical subunits. Each monomer has two types of copper atomic active centers; one behaves as a catalytic center (T2Cu) facilitating $^*NO_2^-$ adsorption/association and the other acts as an electron/proton donating center (T1Cu) promoting the breaking of the N-O bond. According to the mechanism of nitrite reduction on T2Cu and T1Cu, one can rationally speculate that moderate affinity for NO_3^- , protons availability and electrons provision are the key factors for effectively reducing NO_3^- (NO_3^- RR) to NH_3 . Based on the functioning mechanism of Cu-NIRs, we have designed and vs. RHE prepared a Cu-based bimetallic catalyst that is able to mimic the behavior of the two catalytic centers of the Cu-NIRs: herein Cu would act as T2Cu and a second metal would act as T1Cu providing the protons and electrons. Co was selected as the second metal due to his activity toward hydrogen evolution reaction with a high hydrogen adsorption energy (*H). Electrochemical *in-situ* FTIR experiments have demonstrated, that protons adsorbed preferentially on Co in $Cu_{50}Co_{50}$ catalysts (Fig. 4e-g in the manuscript), indicating that Co provides the protons for the NO_3^- reduction. Besides, XPS and Extended X-ray absorption fine structure (XANES) spectra, highlighted electrons transfer from Co species to Cu species. The “enzyme-like” of our catalyst is based on its behavior *i.e.* cooperation between the two active centers in the same way as the ones in the Cu-NIRs.

We have added some descriptions in the “Abstract” to clarify how to achieve the enzyme-inspired design. In addition, the introduction section was well rewritten to clearly explain how we designed the catalysts to obtain a high catalytic ability for NO_3^- RR.

Comment 2:

From nitrate to ammonia or nitrogen, the NO^* is a key intermediate product to determine the final products and the catalytic selectivity but there is no investigation or mention of it.

Answer 2:

Many references have reported that the NO^* is the transfer station in the process of NO_3^- reduction to a variety of products, *e.g.*, NH_3 , N_2 and N_2O . The different abilities to reduce NO^* on different catalysts will produce different products. In this work, NH_3 and NO_2^- were the leading products;

other N-containing products were not detected (Fig. R2). The NO* band was not observed in the previous *in-situ* FTIR spectra because the signal was totally overlapped by the bending vibration signal of H₂O in the region of 1500 cm⁻¹ ~ 1800 cm⁻¹. Here, we used D₂O as the solvent and found two weak vibration bands of adsorbed NO in two different adsorption modes (“bridge” and “on top”) at 1557cm⁻¹ and 1639cm⁻¹ on Cu₅₀Co₅₀ catalyst in the spectra of ATR in-situ FTIR (Fig. R14a). These two peaks on Cu and Co catalysts were weak and difficult to clarify, which might due to the weak adsorption of NO. Thus, it doesn’t merit addition discussion here.

The LSV analysis was also performed to compare the activity of NO reduction on Cu₅₀Co₅₀, Cu and Co catalysts. As shown in Fig. R15, the Cu₅₀Co₅₀ catalyst exhibited a higher current density of NO reduction than Cu and Co catalysts, indicating a higher hydrogenation reduction activity of NO over Cu₅₀Co₅₀ than the pure metallic catalysts. Besides, NH₃ was the only product of NO reduction detected on all these catalysts.

Fig. R14: ATR-FTIR spectra on Cu₅₀Co₅₀ (a), Cu (b) and Co (c) in electrolyte of 100 mM KNO₃ + 1 M KOH with D₂O as the solvent.

Fig. R15: *j*-*E* curve on Cu₅₀Co₅₀, Cu, and Co modified Ni foams in 1 M KOH solution with saturated NO gas (solid lines) or without (dotted line) at a scan rate of 5 mV s⁻¹.

Comment 3:

Please provide sufficient results in terms of the stability data how change or maintain the morphology, chemical compositions, and oxidation states with several repeated cycles.

Answer 3:

Indeed, the durability of the catalysts is another important indicator of catalytic performance. Ten cycle stability tests were performed on $\text{Cu}_{50}\text{Co}_{50}$ in an H-type cell (Fig. R16). The Faraday efficiency remained stable and were more than 90%. Although the $\text{Yield}_{\text{NH}_3}$ decreased gradually to half, which was also two times larger than the Haber–Bosch process (the red dash line in Fig. R16). According to SEM analysis, the decay of the $\text{Yield}_{\text{NH}_3}$ might be due to the nanosheet agglomeration after the consecutive recycling tests (the red cycle in Fig. R17b). XRD (Fig. R17c) and XPS analysis (Fig. R17d and e) were also done on the samples after the consecutive recycling tests. The results demonstrated negligible changes of the chemical compositions and oxidation states, which further confirmed the good stability of the catalyst.

Fig. R16: FE and Yield rate of NH_3 on $\text{Cu}_{50}\text{Co}_{50}/\text{Ni}$ foam under the applied potential of -0.2 V vs. RHE during 10 periods of 1 h electrocatalytic NO_3^- RR. (The red dash line was the $\text{Yield}_{\text{NH}_3}$ of Haber-Bosch process.)

Fig. R17: SEM images of the $\text{Cu}_{50}\text{Ni}_{50}$ catalyst before (a) and after 10 periods of 1 h (b) electrocatalytic NO_3^- RR. XRD spectra (c) and XPS peaks spectra of Cu 2p (d) and Co 2p (e) of $\text{Cu}_{50}\text{Co}_{50}$ catalyst before and after 1 h, 10 periods of 1 h NO_3^- RR operation at -0.2 V vs. RHE.

Comment 4:

The author insisted the Ni foam is inactivity toward nitrate reduction by providing LSV curves collected by the current density in terms of ECSA in Figure S4. Please provide the current density by geometric area and how much the concentrations of nitrate ions could be reduced before and after nitrate reduction tests by using only Ni foam.

Answer 4:

Your suggestion would improve the quality of the manuscript. The LSV curves collected by current density in terms of geometric area and ECSA were displayed below (Fig. R18). A significantly higher current density was observed on $\text{Cu}_{50}\text{Co}_{50}/\text{Ni}$ foam than that on Ni foam. In addition, the slight difference in current density on Ni foam in the electrolyte with and without NO_3^- indicated a low catalytic activity of NO_3^- reduction on Ni foam.⁶ The constant-potential electrolysis of NO_3^- RR was then performed on Ni foam, compared to $\text{Cu}_{50}\text{Co}_{50}/\text{Ni}$ foam. The removal efficiency of nitrate ions ($R_{\text{NO}_3^-}$) was obtained according to Eq. (15)

$$R_{\text{NO}_3^-} = \frac{C_{\text{NO}_3^-}^0 - C_{\text{NO}_3^-}}{C_{\text{NO}_3^-}^0} \times 100\% \quad (15)$$

Where $C_{\text{NO}_3^-}^0$ is the initial concentration of NO_3^- , mmol L^{-1} ; and $C_{\text{NO}_3^-}$ is the NO_3^- residue after electrolysis, mmol L^{-1} .

As shown in Fig. R19, the $R_{\text{NO}_3^-}$ obtained on Ni foam were very low ($< 5\%$), far from that on $\text{Cu}_{50}\text{Co}_{50}/\text{Ni}$ foam (27~66%). Therefore, we consider the Ni foam relatively inactive toward NO_3^-

reduction.

Fig. R18: Current density normalized by geometric area (a) or electrochemical active surface area ($j_{(ECSA)}$)(b)- Electrode potential (E) curve (80% iR corrected) on Ni foam and $\text{Cu}_{50}\text{Co}_{50}$ / Ni foam in 1 M KOH solution containing 100 mM KNO_3 (solid lines) or in the absence of KNO_3 (dotted line) at a sweep rate of 1 mV s^{-1} .

Fig. R19: The removal efficiency of NO_3^- at different electrode potential on $\text{Cu}_{50}\text{Co}_{50}$ / Ni foam and Ni foam in 100 mM KNO_3 + 1 M KOH electrolyte.

Comment 5:

How precisely control the atomic percent between Cu and Co in this electrodeposition method? Also, please provide all XRD data for $\text{Cu}_{70}\text{Co}_{30}$, $\text{Cu}_{50}\text{Co}_{50}$, and $\text{Cu}_{20}\text{Co}_{80}$.

Answer 5:

In order to obtain a close reduction rate of Cu^{2+} and Co^{2+} in the electrodeposition process, in this work, we firstly have added sufficient trisodium citrate to reduce their reduction potential and help

them prone to be co-deposition. Then, the atomic ratio between Cu and Co was regulated by tuning the feeding ratio of CuSO_4 and CoSO_4 . ICP-OES analysis was used to check the accurate contents of Co and Cu in the Cu_xCo_y ($0 < x, y < 100$) catalysts. The Cu_xCo_y catalysts with Cu:Co ratios of around 65:35, 50:50 and 15:85, denoted as $\text{Cu}_{65}\text{Co}_{35}$, $\text{Cu}_{50}\text{Co}_{50}$ and $\text{Cu}_{15}\text{Co}_{85}$, were obtained in the electroplating solutions with the ratio of CuSO_4 to CoSO_4 as 60:40, 45:55 and 10:90, respectively, at the presence of trisodium citrate pentahydrate. The results were shown in Table R11. The XRD data for $\text{Cu}_{65}\text{Co}_{35}$, $\text{Cu}_{50}\text{Co}_{50}$ and $\text{Cu}_{15}\text{Co}_{85}$ were also added to the supporting information (Fig. R20).

Table R11: ICP-OES analysis results for different Cu/Co ratio.

Catalyst	ICP
$\text{Cu}_{65}\text{Co}_{35}$	66:34
$\text{Cu}_{50}\text{Co}_{50}$	51:49
$\text{Cu}_{15}\text{Co}_{85}$	14:86

Fig. R20: XRD spectra of Cu, $\text{Cu}_{65}\text{Co}_{35}$, $\text{Cu}_{50}\text{Co}_{50}$, $\text{Cu}_{15}\text{Co}_{85}$ and Co.

Comment 6:

There is a typo in abstract “The $\text{Cu}_{50}\text{Co}_{50}$ nanosheet delivers a lowset overpotential~”, revise from “lowset” to “lowest”.

Answer 6:

Thank you for careful review. The typo in abstract “lowset” was corrected as “lowest” and marked in red and we also have carefully checked the whole essay to exclude grammar and spelling mistakes.

References

- 1 Su, Y., Muller, K. R., Yoshihara-Saint, H., Najm, I. & Jassby, D. Nitrate Removal in an Electrically Charged Granular-Activated Carbon Column. *Environmental Science & Technology* **55**, 16597-16606 (2021).
- 2 Li, J. *et al.* Efficient Ammonia Electrosynthesis from Nitrate on Strained Ruthenium Nanoclusters. *J. Am. Chem. Soc.* **142**, 7036-7046 (2020).
- 3 Wang, Y. *et al.* Enhanced Nitrate-to-Ammonia Activity on Copper-Nickel Alloys via Tuning of Intermediate Adsorption. *J. Am. Chem. Soc.* **142**, 5702-5708 (2020).
- 4 Yuan, C. *et al.* Growth of ultrathin mesoporous Co₃O₄ nanosheet arrays on Ni foam for high-performance electrochemical capacitors. *Energy Environ. Sci.* **5** (2012).
- 5 Zhang, G. Q., Wu, H. B., Hoster, H. E., Chan-Park, M. B. & Lou, X. W. Single-crystalline NiCo₂O₄ nanoneedle arrays grown on conductive substrates as binder-free electrodes for high-performance supercapacitors. *Energy Environ. Sci.* **5** (2012).
- 6 Zhang, Y. *et al.* Exhaustive Conversion of Inorganic Nitrogen to Nitrogen Gas Based on a Photoelectro-Chlorine Cycle Reaction and a Highly Selective Nitrogen Gas Generation Cathode. *Environ Sci Technol* **52**, 1413-1420 (2018).
- 7 Zhang, Y. *et al.* Electrochemical reduction of nitrate via Cu/Ni composite cathode paired with Ir-Ru/Ti anode: High efficiency and N₂ selectivity. *Electrochim. Acta* **291**, 151-160 (2018).
- 8 Chen, G.-F. *et al.* Electrochemical reduction of nitrate to ammonia via direct eight-electron transfer using a copper-molecular solid catalyst. *Nat. Energy* **5**, 605-613 (2020).
- 9 Prieto, G. *et al.* Design and synthesis of copper-cobalt catalysts for the selective conversion of synthesis gas to ethanol and higher alcohols. *Angew. Chem. Int. Ed. Engl.* **53**, 6397-6401 (2014).
- 10 Hu, T., Wang, C., Wang, M., Li, C. M. & Guo, C. Theoretical Insights into Superior Nitrate Reduction to Ammonia Performance of Copper Catalysts. *ACS Catal.* **11**, 14417-14427 (2021).
- 11 Liu, J.-X., Richards, D., Singh, N. & Goldsmith, B. R. Activity and Selectivity Trends in Electrocatalytic Nitrate Reduction on Transition Metals. *ACS Catal.* **9**, 7052-7064 (2019).
- 12 Nørskov, J. K. *et al.* Origin of the Overpotential for Oxygen Reduction at a Fuel-Cell Cathode. *J. Chem. Phys. B* **108**, 17886-17892 (2004).
- 13 Lange's handbook of chemistry, twelfth edition. Edited by John A. Dean. McGraw-Hill, New York, NY 10020. 1978. 1470 pp. 15 × 23 cm. price \$28.50. *Journal of Pharmaceutical Sciences* **68**, 805-806 (1979).
- 14 Lu, X., Song, H., Cai, J. & Lu, S. Recent development of electrochemical nitrate reduction to ammonia: A mini review. *Electrochem. commun.* (2021).
- 15 Hsieh, H. H. *et al.* Electronic structure of Ni-Cu alloys: The d-electron charge distribution. *Phys. Rev. B* **57**, 15204-15210 (1998).
- 16 Wang, C. *et al.* Regulation of d-Band Electrons to Enhance the Activity of Co-Based Non-Noble Bimetal Catalysts for Hydrolysis of Ammonia Borane. *ACS Sustain. Chem. Eng.* **8**, 8256-8266 (2020).
- 17 Bhattacharjee, S., Waghmare, U. V. & Lee, S. C. An improved d-band model of the catalytic activity of magnetic transition metal surfaces. *Sci. Rep.* **6**, 35916 (2016).
- 18 Ma, F. *et al.* Graphene-like Two-Dimensional Ionic Boron with Double Dirac Cones at Ambient Condition. *Nano Lett.* **16**, 3022-3028 (2016).

REVIEWER COMMENTS

Reviewer #1 (Remarks to the Author):

The manuscript has been revised according to the comments and all my concerns have been suitably addressed. I believe this manuscript now is ready for publication.

Reviewer #2 (Remarks to the Author):

I thank the authors for their responses to the reviewer comments. The manuscript is improved, but I still have the following items to be addressed pertaining to the revised manuscript. (R2A1 refers to reviewer 2, answer 1, etc.)

R2A1: I still struggle with calling this “enzyme-inspired” in a general sense, as the mechanism is different from the hallmark behaviors of enzymes. Enzymes work through a specificity of binding sites; there is not evidence here of any binding specificity. It would be better to restate here that you are focusing on the bifunctional nature of this specific reductase enzyme and drawing parallels between this and your bifunctional catalyst. (Perhaps “inspired by the bifunctional nature of nitrate reductase” would be more appropriate for the text than “enzyme-inspired”.) The actual mechanism here is not really “enzyme-inspired” in the general sense of enzyme functions, but is instead similar to the mechanism used by a specific bifunctional enzyme. I fear there are broader implications to the phrase “enzyme-inspired” that are not substantiated in this manuscript and thus could be misleading.

R2A2: If the authors agree that it is not appropriate to compare this catalyst with the Haber-Bosch method, then I would suggest they either (i) remove this comparison from the main text, or (ii) add qualifiers to this comparison as to why it is not appropriate. I am somewhat confused as to why the comparisons were not changed if the authors agree they are not appropriate.

R2A3: These are interesting points, though they seem to not be incorporated into the revised manuscript. Is there a reason you choose to not incorporate them in the revised manuscript? It would be useful to clarify to the extent to which the Ni foam is essential.

R2A4: I apologize for the lack of clarity in my suggestions for the unit cell dimensions. The Cu(111) slab employed would most typically be referred to as a “4 x 4 surface unit cell”, which is what I was referring to rather than explicit lengths of each unit cell.

R2A5: Can you please provide a reference for stating that the entropy of adsorbates is negligible? It is relatively common practice to include these when calculating free energies of surface reactions.

R2A6: I perhaps should have clarified the intended energies that should be provided. Table S7 is not particularly useful, as these are with respect to an arbitrary reference. The energies with respect to a clearly-stated reference should be provided. (Perhaps the easiest example would be the relevant closed-shell species and a clean slab.)

General comments:

There are still consequential typos in this manuscript. In the abstract, the authors discuss nitrate reductases: I assume they mean nitrite reductase? On page 27, the authors state that “Enthalpic contribution of gas corrections for H₂O, H₂, and NH₃...” I believe they mean entropic contributions.

It is a bit difficult to navigate the revised documents, as there are instances where the authors note changes that are “quoted below”, but the text does not match the text actually input into the manuscript documents. I am not certain if there is anything of substance that is changed, but it makes review more challenging. I encourage the authors to be more careful with this.

Reviewer #3 (Remarks to the Author):

The authors claimed properly and now it is acceptable for publish.

Responds to Reviewers

Collectively, we would like to thank all the reviewers for their careful considerations of our paper. We are delighted to have received such positive feedbacks and feel that the suggested amendments strengthen the paper and its conclusions.

We now provide responses/comments for each point/suggestion raised by reviewer #2. Each of the reviewer's comments are listed below in **blue**; our responses are written in black. In many cases, the reviewer's comments have led to changes in the manuscript and/or electronic supplementary information. For convenience, we highlighted all revisions using **red color** in the revised manuscript and the Supporting Information.

Reviewer #2 (Remarks to the Author).

Comment 1 (R2A1): I still struggle with calling this “enzyme-inspired” in a general sense, as the mechanism is different from the hallmark behaviors of enzymes. Enzymes work through a specificity of binding sites; there is not evidence here of any binding specificity. It would be better to restate here that you are focusing on the bifunctional nature of this specific reductase enzyme and drawing parallels between this and your bifunctional catalyst. (Perhaps “inspired by the bifunctional nature of nitrate reductase” would be more appropriate for the text than “enzyme-inspired”.) The actual mechanism here is not really “enzyme-inspired” in the general sense of enzyme functions, but is instead similar to the mechanism used by a specific bifunctional enzyme. I fear there are broader implications to the phrase “enzyme-inspired” that are not substantiated in this manuscript and thus could be misleading.

Answer 1:

Thanks for your kind comment. We agree that it would be more appropriate and precise to identify the studied CuCo catalysts inspired by the bifunctional nature of nitrite reductase for NO_3^- RR. After reconsideration, we rephrased the title “Enzyme-Inspired CuCo Nanosheet for Electrochemical Synthesis of Ammonia at an Ampere-Level Current Density” as “**Ampere-Level Current Density Ammonia Electrochemical Synthesis using CuCo nanosheets simulating Nitrite reductase bifunctional Nature**”. The sentence “Herein, we present a CuCo bimetallic catalyst able to simulate the behavior of the nitrate reductase active centers.” in the abstract was rewritten as “**Herein, we present a CuCo bimetallic catalyst able to imitate the bifunctional nature of nitrite reductase.**” to avoid any misleading messages. Meanwhile, the sentence “These encouraging results highlighted that the enzyme-inspired design is a promising cost-effective way to obtain high-performance

electrocatalysts for NH₃ production and environmental remediation.” in the abstract was reworded as “These encouraging results highlighted that designing catalysts based on the bifunctional nature of specific enzymes is a promising cost-effective way to obtain high-performance electrocatalysts for NH₃ production and environmental remediation.”. All these changes made are to avoid any misleading messages

Comment 2 (R2A2): If the authors agree that it is not appropriate to compare this catalyst with the Haber-Bosch method, then I would suggest they either (i) remove this comparison from the main text, or (ii) add qualifiers to this comparison as to why it is not appropriate. I am somewhat confused as to why the comparisons were not changed if the authors agree they are not appropriate.

Answer 2:

Thanks for your kind suggestion. It is correct that in the present study the electrochemical synthesis of NH₃ from NO₃⁻ was for now performed at a lab scale, while the Haber-Bosch method is a well-established industrial process. However, several published articles performed a technico-economical comparisons between the two processes¹⁻⁴. Worth of the existing literature, herein we decided to list the production rate of NH₃ formation from NO₃⁻RR and the Haber-Bosch method to emphasize the advantages of our system. However, to avoid any confusion, some features were rewritten.

The sentence “The NH₃ production rate reaches 4.8 mmol cm⁻² h⁻¹ (960 mmol g_{cat}⁻¹ h⁻¹) which is about 5 times higher than the Haber-Bosch route (200 mmol g_{cat}⁻¹ h⁻¹).” in the abstract (line 11) was rewritten as “The NH₃ production rate reaches a high activity of 4.8 mmol cm⁻² h⁻¹ (960 mmol g_{cat}⁻¹ h⁻¹)”.

The word “alternative” on page 2 was changed to “complementary” since we agree that the current electrosynthesis of NH₃ is at a lab-scale, and further efforts are needed to reach an industrial scale-up.

The sentence “Based on the charge consumed during the CuCo electrodeposition on Ni foam substrate, the highest mass activity of NH₃ yield (Yield_{mass-NH₃}) on Cu₅₀Co₅₀ was estimated roughly to 960 mmol g_{cat}⁻¹ h⁻¹ which is around 5 times larger than the Haber-Bosch process (200 mmol g_{cat}⁻¹ h⁻¹).” in the second paragraph, page 10, was rephrased as “Based on the charge consumed during the CuCo electrodeposition on Ni foam substrate, the mass activity of NH₃ yield (Yield_{mass-NH₃}) on Cu₅₀Co₅₀ was estimated roughly to 960 mmol g_{cat}⁻¹ h⁻¹ at 0.2 V. The obtained Yield_{mass-NH₃} was slightly underestimated since hydrogen evolution was observed during CuCo electrodeposition. It should be noted that the production of NH₃ from electrocatalytic NO₃⁻RR is currently at a laboratory scale. More follow-up pilot tests and scale-up work are required to meet the industrial demands. Based on the preliminary calculations we consider that our work inspired by the bifunctional nature of nitrite reductase, provides a new expectation and shows a great prospect, and after further development could compete with the well-established Haber-Bosch process^{12,32} which currently shows a Yield_{mass-NH₃} of ca. 200 mmol g_{cat}⁻¹ h⁻¹ at industrial scale.”

The previous description “Though the $\text{Yield}_{\text{NH}_3}$ decreased gradually by half of its value, it was still over two times larger than the Haber–Bosch process.” was deleted.

The sentence “This process exhibited an activity 5 times higher than the Haber–Bosch process and can be proposed as a sustainable and eco-friendly alternative.” in the conclusions was rewritten as “This process exhibited a high activity and could be proposed as a sustainable and eco-friendly complementary route of NH_3 production.”

Comment 3 (R2A3): These are interesting points, though they seem to not be incorporated into the revised manuscript. Is there a reason you choose to not incorporate them in the revised manuscript? It would be useful to clarify to the extent to which the Ni foam is essential.

Answer 3:

We acknowledge your valuable comments. We have added the sentence, “Ni foam is widely used as a supporting substrate for nanostructured electrocatalysts due to its smooth surface and good conductivity, benefiting the electrodeposition by an efficient electron transfer¹⁵⁻¹⁷. Meanwhile, Ni was proved as a relatively inert material for $\text{NO}_3\text{-RR}$ ^{18,19}, without affecting the CuCo catalysts’ performance.” in the first paragraph of “Preparation and characterization of CuCo bimetallic electrocatalysts” section (page 4), to emphasize the importance of using Ni foam as the substrate for electrodeposition of CuCo catalysts.

Comment 4 (R2A4): I apologize for the lack of clarity in my suggestions for the unit cell dimensions. The Cu(111) slab employed would most typically be referred to as a “4 x 4 surface unit cell”, which is what I was referring to rather than explicit lengths of each unit cell.

Answer 4:

Thank you for your careful review. We accepted your suggestion and provided a more distinct description for the used unit cell. The description “The converged unit cell models of Cu ($3.64 \text{ \AA} \times 3.64 \text{ \AA} \times 3.64 \text{ \AA}$), Co ($3.52 \text{ \AA} \times 3.52 \text{ \AA} \times 3.52 \text{ \AA}$) and CuCo ($3.78 \text{ \AA} \times 3.49 \text{ \AA} \times 3.49 \text{ \AA}$) were used in DFT calculations, respectively. The dimension of supercell of Cu (111) ($8.91 \text{ \AA} \times 10.28 \text{ \AA} \times 24.20 \text{ \AA}$), Co (111) ($8.62 \text{ \AA} \times 9.95 \text{ \AA} \times 19.06 \text{ \AA}$) and CuCo (111) ($9.03 \text{ \AA} \times 9.87 \text{ \AA} \times 19.13 \text{ \AA}$) were used, respectively. These supercells were constructed and contained three layers and a sufficient vacuum layer of 15 \AA thicknesses. The bottom two layers were fixed and the top layer was fully relaxed.” in the method section of page 24 in the manuscript was rewritten as “The converged unit cell models of Cu ($3.64 \times 3.64 \times 3.64 \text{ \AA}^3$), Co ($3.52 \times 3.52 \times 3.52 \text{ \AA}^3$) and CuCo ($3.78 \times 3.49 \times 3.49 \text{ \AA}^3$) were used in DFT calculations, respectively. The dimension of a 2×2 supercell of Cu (111) ($8.91 \times 10.28 \text{ \AA}^2$), a 2×2 supercell of Co (111) ($8.62 \times 9.95 \text{ \AA}^2$) and a 2×2 supercell of CuCo (111) ($9.03 \times 9.87 \text{ \AA}^2$) were used, respectively. These supercells were constructed and contained three layers and a sufficient vacuum layer of 15 \AA thicknesses. For the structural optimization, the bottom two layers were fixed and the top layer was fully relaxed⁶⁹.”

Comment 5 (R2A5): Can you please provide a reference for stating that the entropy of adsorbates is negligible? It is relatively common practice to include these when calculating free energies of surface reactions.

Answer 5:

Thanks for your kind guide. The entropy of adsorbates was calculated and accounted for the calculation of free energies of the NO₃⁻RR. The whole slab atoms were fixed during the calculation for the vibration analysis of adsorbates. The difference of ΔZPE between the slab whole fixed and the slab fixed bottom two layers used before was subtle (Table R1), and we updated the data of zero-point energies (ZPE) shown in **Supplementary Table S7** (here in Table R2).

The correction of entropy (TS), calculated electronic energies (E) of the intermediates involved in NO₃⁻RR and the calculated Gibbs free energies (ΔG) on different catalysts surfaces were listed in Table R3, R4, and R5, respectively. Fig. 6 of Gibbs free energies for different intermediates of NO₃⁻RR and HER in the manuscript, Page 19, is updated and shown below (**Figure R1**).

Table R1: The ΔZPE for the elementary steps involved in NO₃⁻RR on different catalysts' surfaces (in eV), with all the slab atoms fixed during the vibrational analyses (used now) or the bottom two layers fixed and the top layer fully relaxed (used before), respectively.

ΔZPE	Cu(111)		Co(111)		CuCo(111)	
	All the slab atoms fixed	Bottom two layers fixed	All the slab atoms fixed	Bottom two layers fixed	All the slab atoms fixed	Bottom two layers fixed
ZPE(*NO ₃)-ZPE(*)	0.39	0.40	0.40	0.40	0.40	0.40
ZPE(*NO ₂)-ZPE(*NO ₃)	-0.13	-0.13	-0.13	-0.11	-0.14	-0.14
ZPE(*NO)-ZPE(*NO ₂)	-0.10	-0.10	-0.10	-0.13	-0.07	-0.07
ZPE(*NOH)-ZPE(*NO)	0.30	0.30	0.28	0.28	0.27	0.29
ZPE(*NHOH)-ZPE(*NOH)	0.32	0.32	0.32	0.33	0.31	0.30
ZPE(*NH ₂ OH)-ZPE(*NHOH)	0.32	0.31	0.34	0.34	0.33	0.32
ZPE(*NH ₂)-ZPE(*NH ₂ OH)	-0.43	-0.42	-0.43	-0.43	-0.43	-0.42
ZPE(*NH ₃)-ZPE(*NH ₂)	0.32	0.31	0.33	0.33	0.34	0.33
ZPE(*)-ZPE(*NH ₃)	-0.99	-0.99	-1.01	-1.01	-1.01	-1.01

Table R2: The correction of zero-point energy (ZPE) of adsorption species on different catalysts' surfaces (in eV). All the slab atoms were fixed during the vibrational analyses. (The corresponding table in the Supporting Information is Supplementary Table S7 and marked in red.)

	Cu(111)	Co(111)	CuCo(111)
*NO ₃	0.39	0.40	0.40
*NO ₂	0.26	0.27	0.26
*NO	0.16	0.17	0.19
*NOH	0.46	0.45	0.46
*NHOH	0.78	0.77	0.77
*NH ₂ OH	1.10	1.11	1.10
*NH ₂	0.67	0.68	0.67
*NH ₃	0.99	1.01	1.01
*H	0.17	0.17	0.18

Table R3: The correction of entropy (TS) of adsorption species on different catalysts' surfaces (in eV). All the slab atoms were fixed during the vibrational analyses. (T = 298.15 K). (The corresponding table in the Supporting Information is Supplementary Table S8 and marked in red.)

	Cu(111)	Co(111)	CuCo(111)
*NO ₃	0.23	0.27	0.26
*NO ₂	0.24	0.20	0.22
*NO	0.14	0.12	0.17
*NOH	0.16	0.16	0.15
*NHOH	0.15	0.14	0.15
*NH ₂ OH	0.21	0.19	0.25
*NH ₂	0.10	0.09	0.09
*NH ₃	0.14	0.17	0.12
*H	0.00	0.00	0.00

Table R4: Calculated electronic energies (E) of adsorption species on different catalysts' surfaces (in eV). (The corresponding table in the Supporting Information is Supplementary Table S9 and marked in red.)

	Cu(111)	Co(111)	CuCo(111)
*	-162.86	-317.95	-233.09
*NO ₃	-188.29	-343.62	-258.94
*NO ₂	-182.66	-338.01	-252.94
*NO	-176.57	-332.66	-247.60
*NOH	-179.97	-335.96	-251.05
*NHOH	-183.84	-339.45	-254.66
*NH ₂ OH	-187.60	-342.90	-258.09
*NH ₂	-178.82	-334.23	-249.18
*NH ₃	-182.92	-338.19	-253.36
*H	-166.57	-321.82	-236.97

Table R5: Calculated Gibbs free energies (ΔG) of adsorption species for NO₃⁻RR on different catalysts' surfaces with respect to the reference of NO₃⁻(l) + * (in eV). (The corresponding table in the Supporting Information is Supplementary Table S10 and marked in red.)

	Cu(111)		Co(111)		CuCo(111)	
	U = 0 V	U = -0.2 V	U = 0 V	U = -0.2 V	U = 0 V	U = -0.2 V
Reference NO ₃ ⁻ (l) + *	0	0	0	0	0	0
ΔG (*NO ₃)	0.41	0.61	0.14	0.34	-0.03	0.17
ΔG (*NO ₂)	-1.50	-1.70	-1.72	-1.92	-1.54	-1.74
ΔG (*NO)	-2.83	-3.43	-3.80	-4.40	-3.63	-4.23
ΔG (*NOH)	-2.49	-3.29	-3.41	-4.21	-3.33	-4.13
ΔG (*NHOH)	-2.58	-3.58	-3.10	-4.10	-3.18	-4.18
ΔG (*NH ₂ OH)	-2.63	-3.83	-2.81	-4.01	-2.93	-4.13
ΔG (*NH ₂)	-5.03	-6.43	-5.34	-6.74	-5.16	-6.56
ΔG (*NH ₃)	-5.39	-6.99	-5.60	-7.20	-5.58	-7.18
ΔG (NH ₃)	-5.42	-7.02	-5.42	-7.02	-5.42	-7.02

Table R6: Calculated Gibbs free energies (ΔG) of adsorption species for HER on different catalysts' surfaces with respect to the reference of $H^+ + e^- + *$ (in eV). (The corresponding table in the Supporting Information is Supplementary Table S11 and marked in red.)

	Cu(111)		Co(111)		CuCo(111)	
	U = 0 V	U = -0.2 V	U = 0 V	U = -0.2 V	U = 0 V	U = -0.2 V
Reference $H^+ + e^- + *$	0	0	0	0	0	0
$\Delta G(*H)$	-0.08	-0.28	-0.25	-0.45	-0.25	-0.45

Figure R1: DFT calculations of NO_3^- RR and HER on Cu(111), Co(111) and CuCo(111). Reaction free energies for different intermediates of NO_3^- RR (a) and HER (b) at -0.2 V vs. RHE on CuCo(111), pure Cu(111), and pure Co(111) surfaces, respectively. The corresponding table in the main text is Fig. 6 and marked in red.

Comment 6 (R2A6): I perhaps should have clarified the intended energies that should be provided. Table S7 is not particularly useful, as these are with respect to an arbitrary reference. The energies with respect to a clearly-stated reference should be provided. (Perhaps the easiest example would be the relevant closed-shell species and a clean slab.)

Answer 6:

We appreciate your helpful questions and suggestion again. We should apologize for any misunderstanding caused by the unclear reference. $NO_3^-(l) + *$ and $H^+ + e^- + *$ were regarded as the zero-point reference of Gibbs free energies for NO_3^- RR and HER, respectively. (* is the active site of a clean slab) As the energy of negatively charged species like NO_3^- and OH^- are difficult to treat in the thermodynamic calculation; instead, they can be approached using the stable molecules HNO_3 and H_2 in the gas phase ($HNO_{3(g)}$ and $H_{2(g)}$) and liquid H_2O ($H_{2O(l)}$) instead. The $HNO_{3(g)}$, $H_{2O(l)}$

and $H_{2(g)}$ are used as references when calculating the Gibbs free energies of reaction intermediates. The zero-point energies (ZPE) and entropy (TS) (in eV) corrections for Gibbs free energy calculations of $HNO_{3(g)}$, $H_2O_{(l)}$, $NH_{3(g)}$ and $H_{2(g)}$ were obtained through vibrational analyses. The calculated TS were close to the data cited from CRC handbook of chemistry and physics⁵, as shown in Table R7.

Table R7: The electronic energies (E), zero-point energies (ZPE) and entropy (TS) corrections for $H_2O_{(l)}$, $H_{2(g)}$, $NH_{3(g)}$ and $HNO_{3(g)}$ used in the Gibbs free energy calculations. The data marked in bold are cited from CRC handbook of chemistry and physics⁵. ($T = 298.15$ K, $p_{gas} = 1$ atm). (The corresponding table in the Supporting Information is **Supplementary Table S5** and marked in red.)

Molecule	E / eV	ZPE / eV	TS / eV	TS(CRC handbook) / eV
$H_2O_{(l)}$	-14.22	0.57	0.67	\
$H_{2(g)}$	-6.77	0.27	0.40	0.40
$NH_{3(g)}$	-19.54	0.91	0.60	0.60
$HNO_{3(g)}$	-28.61	0.69	0.83	0.82

To clarify the reference and provide the energies with respect to a clearly-stated reference, we added the description of “**In the thermodynamic calculation, the energy of negatively charged species like NO_3^- and OH^- are difficult to determine. However, they can be approached using the stable molecules HNO_3 and H_2 in the gas phase ($HNO_{3(g)}$ and $H_{2(g)}$) and liquid H_2O ($H_2O_{(l)}$) instead. The $HNO_{3(g)}$, $H_2O_{(l)}$, $H_{2(g)}$ and a clean slab are used as references when calculating the Gibbs free energies of reaction intermediates. The zero-point energies (ZPE) and entropy (TS) (in eV) corrections for Gibbs free energy calculations of $HNO_{3(g)}$, $H_2O_{(l)}$, $NH_{3(g)}$ and $H_{2(g)}$ were obtained through vibrational analyses. The calculated TS were close to the data cited from CRC handbook of chemistry and physics³, as shown in **Supplementary Table S5**.” in Supplementary Note S1 in the Supporting Information (page 34). Furthermore, the Calculated Gibbs free energies (ΔG) of adsorption species for NO_3^-RR (**Supplementary Table S10**) and HER (**Supplementary Table S11**) on different catalysts surfaces were presented with respect to the reference of $NO_3^-(l) + *$ and $H^+ + e^- + *$, respectively.**

We redescribed the calculation of Gibbs free energy for NO_3^-RR in detail, especially the first step of $NO_3^-(l)$ adsorption onto the electrode surface, and added the below descriptions into the Supporting Information and marked them in red.

“**Details of DFT calculations.**

1. The Gibbs free energy of $NO_3^-(l)$ adsorption onto the electrode surface.

At 0 V vs. RHE, $H^+ + e^- \rightleftharpoons \frac{1}{2}H_2$ is in equilibrium at $p_{H_2} = 1$ atm (Eqs. (S1) and (S2)).

$$G(H^+) + G(e^-) = \frac{1}{2}G(H_2) \quad (S1)$$

$$G(OH^-) = G(H_2O) - \frac{1}{2}G(H_2) \quad (S2)$$

The process of NO_3^- adsorbed onto the electrode surface (Eq. (S3)) was divided into three steps (Eqs. (S4-S6)), as shown in Scheme S1^{1,2}.

Supplementary Scheme S1. The thermodynamic cycle used to calculate the Gibbs free energy of NO_3^- in the aqueous phase (NO_3^-) adsorbed onto the electrode surface^{1,2}. The thermodynamic values (0.317 eV and 0.074 eV) are obtained from the CRC handbook of chemistry and physics³.

Firstly, the Gibbs free energy for $HNO_3(l)$ formation from NO_3^- is 0.317 eV (Eq. (S4)). The change in Gibbs free energy of the vaporization of $HNO_3(l)$ was then calculated from the Gibbs free energy difference between the standard formation of HNO_3 in liquid ($HNO_3(l)$) (-0.836 eV) and gas phase ($HNO_3(g)$) (-0.762 eV) and was equal to 0.074 eV (Eq. (S5)). The Gibbs free energy of NO_3^- adsorption on the surface ($*NO_3$) (Eq.(S6)), following the Eq. (S7).

$$\Delta G_{ads}(*NO_3) = G(*NO_3) + \frac{1}{2}G_{gas}(H_2) - G(*) - G_{gas}(HNO_3) \quad (S7)$$

Ultimately, the overall Gibbs free energy change (ΔG_{*NO_3}) for NO_3^- adsorption from the solution phase on to electrode surface ($NO_3^-(l) + * \rightarrow *NO_3 + e^-$) was calculated following Eq. (S8):

$$\begin{aligned} \Delta G_{*NO_3} &= \Delta G_{ads}(*NO_3) + 0.074 \text{ eV} + 0.317 \text{ eV} - eU \\ &= G(*NO_3) + \frac{1}{2}G_{gas}(H_2) - G(*) - G_{gas}(HNO_3) + 0.391 \text{ eV} - eU \\ &= E(*NO_3) + ZPE(*NO_3) - TS(*NO_3) + \frac{1}{2}[(E_{gas}(H_2) + ZPE_{gas}(H_2) - TS_{gas}(H_2))] - \end{aligned}$$

$$E(*) - E_{gas}(HNO_3) - ZPE_{gas}(HNO_3) + TS_{gas}(HNO_3) + 0.391 \text{ eV} - eU \quad (S8)$$

Here we neglected the ZPE and rotational, translational, and vibrational free energy contributions for slab (i.e., $G(*) = E(*)$).¹ The U is the potential at the electrode and e is the transferred charge.

2. The NO₃⁻RR on different catalysts surfaces were simulated according to the following reactions⁴:

Where the * represents the active sites.

For each subsequent reaction, the free energies were given after gas correction, following Eq. (S18):

$$\Delta G = \Delta E + \Delta ZPE - T\Delta S - eU \quad (S18)$$

where ΔE is the energy obtained by the difference between reactant and product, ΔZPE denotes the change of zero-point energy. ΔS is the change in entropy for each reaction. The entropies of adsorbate and adsorption site are negligible. Here, U is the potential at the electrode and e is the transferred charge.”

Comment 7 (General comments):

There are still consequential typos in this manuscript. In the abstract, the authors discuss nitrate reductases: I assume they mean nitrite reductase? On page 27, the authors state that “Enthalpic contribution of gas corrections for H2O, H2, and NH3...” I believe they mean entropic contributions.

It is a bit difficult to navigate the revised documents, as there are instances where the authors note changes that are “quoted below”, but the text does not match the text actually input into the manuscript documents. I am not certain if there is anything of substance that is changed, but it makes review more challenging. I encourage the authors to be more careful with this.

Answer 7:

We have to apologize for the typos in the main text sincerely. We have corrected the mistakes and reviewed the full text again to ensure no typos were left. Considering the limitation of word numbers by the journal, we added the key message into the main text rather than the full description.

(1) We added the sentence “**Dipole corrections in the z direction were included in all computations to minimize inaccuracies in the total energy because of simulated slab interactions.**” In the “DFT calculations” section in Methods (page 24) to make sure all the necessary information was included in the manuscript.

(2) Part of the descriptions in the “DFT calculations” section in the main text was deleted and displayed in the section of “Details of DFT calculations” in Supporting Information (page 4) and marked in red.

(3) The initial data calculated for the Gibbs free energy calculations was described in the Supplementary Note S2 in Supporting Information (Page 34) and marked in red.

(4) The rationality of the CuCo bimetallic model used in this work, which was previously thoroughly discussed in the **Respond to Reviewers Letter** but not appearing in the main text or Supporting Information, was currently added in the Supplementary Note S3 in Supporting Information (Page 34) and marked in red.

Finally, we hope that the additions we have made to manuscript are sufficient for the reviewer to deem that this work merits publication as an article in Nature communications. We are somewhat limited by the word count and thus, have attempted to address these concerns as concisely as possible to produce an improved manuscript.

References :

- R1 Li, J. *et al.* Efficient Ammonia Electrosynthesis from Nitrate on Strained Ruthenium Nanoclusters. *J. Am. Chem. Soc.* **142**, 7036-7046 (2020).
- R2 He, W. *et al.* Splicing the active phases of copper/cobalt-based catalysts achieves high-rate tandem electroreduction of nitrate to ammonia. *Nature Communications* **13**, 1-13 (2022).
- R3 Liu, H. *et al.* Efficient Electrochemical Nitrate Reduction to Ammonia with Copper-Supported Rhodium Cluster and Single-Atom Catalysts. *Angewandte Chemie International Edition* **61**, e202202556 (2022).
- R4 van Langevelde, P. H., Katsounaros, I. & Koper, M. T. M. Electrocatalytic Nitrate Reduction for Sustainable Ammonia Production. *Joule* **5**, 290-294 (2021).
- R5 Lide, D. R. *CRC Handbook of Chemistry and Physics*. (CRC Press, Boca Raton, FL, 2009).

REVIEWERS' COMMENTS

Reviewer #2 (Remarks to the Author):

The authors have satisfactorily addressed my comments, and the manuscript is ready to be published.